# Exploratory association between multimodal AI-derived digital biomarkers and in-hospital mortality in adult patients with pneumonia: A proof-of-concept study

Alejandro Hernández-Arango[1,2*], Daniel Mejía Arrieta[3], Christian Andrés Díaz León[4], Juan G. Paniagua Castrillon[5], Julián Rondón-Carvajal[6], Melissa Alejandra Acosta[7], David Restrepo[7], Wayner Barrios[8], Santiago Álvarez-López[9], Jesús Francisco Vargas-Bonilla[10], Hernán Felipe García Arias[10], José Julián Garcés Echeverri[11], Carlos Salazar-Martínez[11], Olga Lucia Quintero Montoya[4]

**1** Department of Internal Medicine, School of Medicine, University of Antioquia, Medellín, Colombia, **2** Hospital Alma Máter de Antioquia, Medellín, Colombia, **3** Department of Radiology, School of Medicine, University of Antioquia, Medellín, Colombia, **4** Computation and Analytics Area, School of Applied Sciences and Analytics, University EAFIT, Medellín, Colombia, **5** Faculty of Exact and Applied Sciences, ITM University Institution, Medellín, Colombia, **6** Pulmonology Program, CardioVID Clinic, CES University, Medellín, Colombia, **7** Pink Technologies Group by Humath, La Estrella, Colombia, **8** Wiqonn Technologies, Barranquilla, Colombia, **9** Hematology Program, National University of Colombia, Bogotá, Colombia, **10** University of Antioquia, Medellín, Colombia, **11** School of Applied Sciences and Analytics, University EAFIT, Medellín, Colombia

\* alejandro.hernandeza@udea.edu.co

## Abstract

Pneumonia remains a leading cause of in-hospital mortality worldwide. Current prognostic tools such as the IDSA/ATS severity score have meaningful limitations, particularly in capturing dynamic disease progression or integrating heterogeneous biological signals. Artificial intelligence (AI) offers the opportunity to derive complementary prognostic information from routinely collected electronic health record data. This exploratory, proof-of-concept retrospective study enrolled adults (≥18 years) admitted with a primary diagnosis of acute pneumonia at Hospital Alma Máter de Antioquia (Medellín, Colombia) between January 1 and June 30, 2024. After applying pre-defined exclusion criteria, 121 patients (19 non-survivors, 15.7%) comprised the final analytic cohort. Three independent AI modules were applied: (i) a ResNet-18 deep learning model quantified lung consolidation from chest radiographs (CXRs) using Class Activation Mapping; (ii) a Spanish regular-expression natural language processing (NLP) pipeline extracted modified IDSA/ATS severity scores from clinical notes; and (iii) NeuroKit2-based quantitative heart rate variability (HRV) analysis processed electrocardiogram (ECG) signals digitised from PDF archives. Bivariate associations with all-cause in-hospital mortality were examined using logistic regression. Several features exhibited statistically significant associations with mortality under conventional thresholds: AI-quantified total lung compromise ratio (OR 8.32, 95% CI

**Data availability statement:** The de-identified feature-level dataset, anonymised raw clinical data, and analysis code supporting the findings of this study are publicly available on Zenodo https://doi.org/10.5281/zenodo.19361102, https://zenodo.org/records/19361102 The deposited archive contains CXR compromise ratios, NLP-derived IDSA/ATS severity indicators, ECG/HRV features for all 121 patients, and anonymised raw clinical records, along with the statistical analysis scripts used to produce the results reported in this manuscript.

**Funding:** This work was supported by the Ministerio de Ciencia, Tecnología e Innovación (Minciencias) of Colombia, call number 895-2021, project code 563-2021, project title 'HuMath Curie: decisiones médicas confiables en Unidades de cuidado Respiratorio a partir de Inteligencia Artificial en Enfermedades pulmonares tipo COVID 19.' The funders had no role in study design, data collection and analysis, decision to publish, or preparation of the manuscript.

**Competing interests:** The authors have declared that no competing interests exist.

1.23–56.29), NLP-derived IDSA/ATS severity score (OR 1.78, 95% CI 1.09–2.88), and a coherent ECG/HRV profile characterised by higher heart rate (120.1 vs. 84.4 bpm, $p = 0.023$), reduced RMSSD (4.1 vs. 23.5 ms, $p = 0.041$), reduced Poincaré SD1 (3.0 vs. 17.6 ms, $p = 0.041$), and T-wave amplitude reductions surviving FDR correction in multiple leads. Given the small sample size and low event count (n = 19; events-per-variable ≈ 4), all associations are preliminary and hypothesis-generating only. These proof-of-concept findings suggest that integrated multimodal AI biomarkers automatically derived from low-resource clinical data can capture a cardiopulmonary stress profile associated with pneumonia mortality, and support the design of larger prospective validation studies.

## Author summary

In our study, we explored whether artificial intelligence tools applied to routinely collected hospital data could help identify patients with pneumonia who are at higher risk of dying during their hospital stay. Current severity scores used for pneumonia have important limitations: they rely on a fixed set of variables and do not capture the dynamic, multisystem nature of the disease. We analysed data from 121 adults admitted with pneumonia to a tertiary hospital in Medellín, Colombia. We used three independent AI modules to automatically extract information from chest radiographs, clinical notes, and electrocardiograms. Several AI-derived features—including lung compromise on imaging, disease severity from clinical notes, and heart rate variability from electrocardiograms—showed statistically significant associations with in-hospital mortality in exploratory bivariate analyses. Because our study was small and exploratory, these findings should be considered hypothesis-generating rather than definitive. Nevertheless, they suggest that multimodal AI biomarkers extracted from low-resource clinical data can capture a cardiopulmonary stress profile associated with pneumonia mortality, supporting the design of larger, prospective validation studies.

## Introduction

Pneumonia remains a leading cause of global morbidity and mortality, particularly impacting vulnerable groups such as the elderly and immunocompromised [1,2]. High mortality rates persist despite therapeutic advances, especially among hospitalised patients and those requiring intensive care unit (ICU) admission [2–4]. This underscores the critical need for accurate prognostic tools in acute care settings to guide essential clinical decisions, including triage, resource allocation, treatment intensity, and goals-of-care discussions [5,6]. Effective risk stratification upon presentation can significantly influence patient outcomes [7].

Established prognostic tools like the Pneumonia Severity Index (PSI) and the CURB-65 score are widely used to predict 30-day mortality but possess notable

limitations. The PSI, while comprehensive, is complex and cumbersome for rapid assessment; the simpler CURB-65 may oversimplify risk and perform suboptimally in specific populations (e.g., immunocompromised) or for predicting outcomes beyond short-term mortality. Crucially, both scores were primarily validated for mortality risk and often fall short in accurately predicting the need for ICU-level interventions [6]. Their reliance on a limited set of static variables measured at admission restricts their ability to capture the dynamic pathophysiology of severe pneumonia [8].

Artificial intelligence (AI) and machine learning (ML) offer promising alternatives, demonstrating the capacity to analyse complex, high-dimensional electronic health record (EHR) datasets and identify intricate prognostic patterns missed by traditional methods [7]. AI/ML models have shown superior performance in predicting mortality and other adverse outcomes in critical illnesses like ARDS [9] compared to conventional scores. A key advantage is their ability to integrate multimodal data—fusing information from diverse sources such as medical imaging, clinical text, and physiological signals—to create a more holistic patient assessment [10].

This study focuses on leveraging AI to integrate digital biomarkers from three key modalities:

- **AI-analysed chest imaging:** Automated analysis of chest radiographs (CXRs) provides objective quantification of lung injury (e.g., lung compromise ratios), which correlates strongly with pneumonia severity and mortality [11,12].

- **Natural language processing (NLP) of clinical text:** NLP techniques can extract critical prognostic information—such as high-risk diagnoses or documented severity assessments (IDSA/ATS criteria)—from unstructured clinical notes [13,14].

- **Quantitative electrocardiogram (ECG) analysis:** Beyond qualitative interpretation, quantitative ECG features and HRV metrics reflect cardiac stress and autonomic function, providing prognostic insights not captured by standard scores [15–17].

Recent AI studies have explored individual modalities for pneumonia prognosis. Deep learning models for CXR analysis have achieved radiologist-level performance in detecting pulmonary opacities [18], and multimodal architectures have proposed late fusion of imaging and clinical features for severity classification [10]. ECG digitisation from paper archives has gained attention as a cost-effective method for retrospective signal analysis in settings where native digital recordings are unavailable [19], while NLP extraction of clinical criteria from electronic health records has primarily been developed for English-language text; Spanish-language clinical NLP pipelines remain largely unexplored [20]. Although these individual components have been studied separately, no prior work has combined AI-derived CXR, ECG, and NLP biomarkers within a single integrated pipeline for pneumonia mortality assessment, particularly in a low- and middle-income country (LMIC) Latin American setting.

This study is, to our knowledge, the first to integrate automatic CXR consolidation quantification, Spanish-language NLP-based clinical severity scoring, and quantitative ECG analysis from digitised paper archives within a unified AI-driven pipeline applied to the same pneumonia cohort. This exploratory, proof-of-concept study therefore aimed to examine associations between novel AI-derived multimodal digital biomarkers and in-hospital mortality among adult pneumonia patients in acute and critical care, using clinical data formats widely available in resource-limited hospitals (CXR images, unstructured Spanish clinical notes, and paper-based ECGs digitised from PDF archives).

## Results

### Cohort characteristics

The study cohort comprised 121 patients, of whom 19 (15.7%) were non-survivors and 102 (84.3%) were survivors (Fig 1). The overall median age was 66.0 years [IQR 56.0–81.0], with no significant difference observed between non-survivors (median 66.0 years) and survivors (median 66.0 years, $p = 0.789$). Females constituted 42.1% of the cohort (36.8% of non-survivors vs. 43.1% of survivors, $p = 0.797$). Sex-specific mortality rates were 13.7% (7/51) for females and

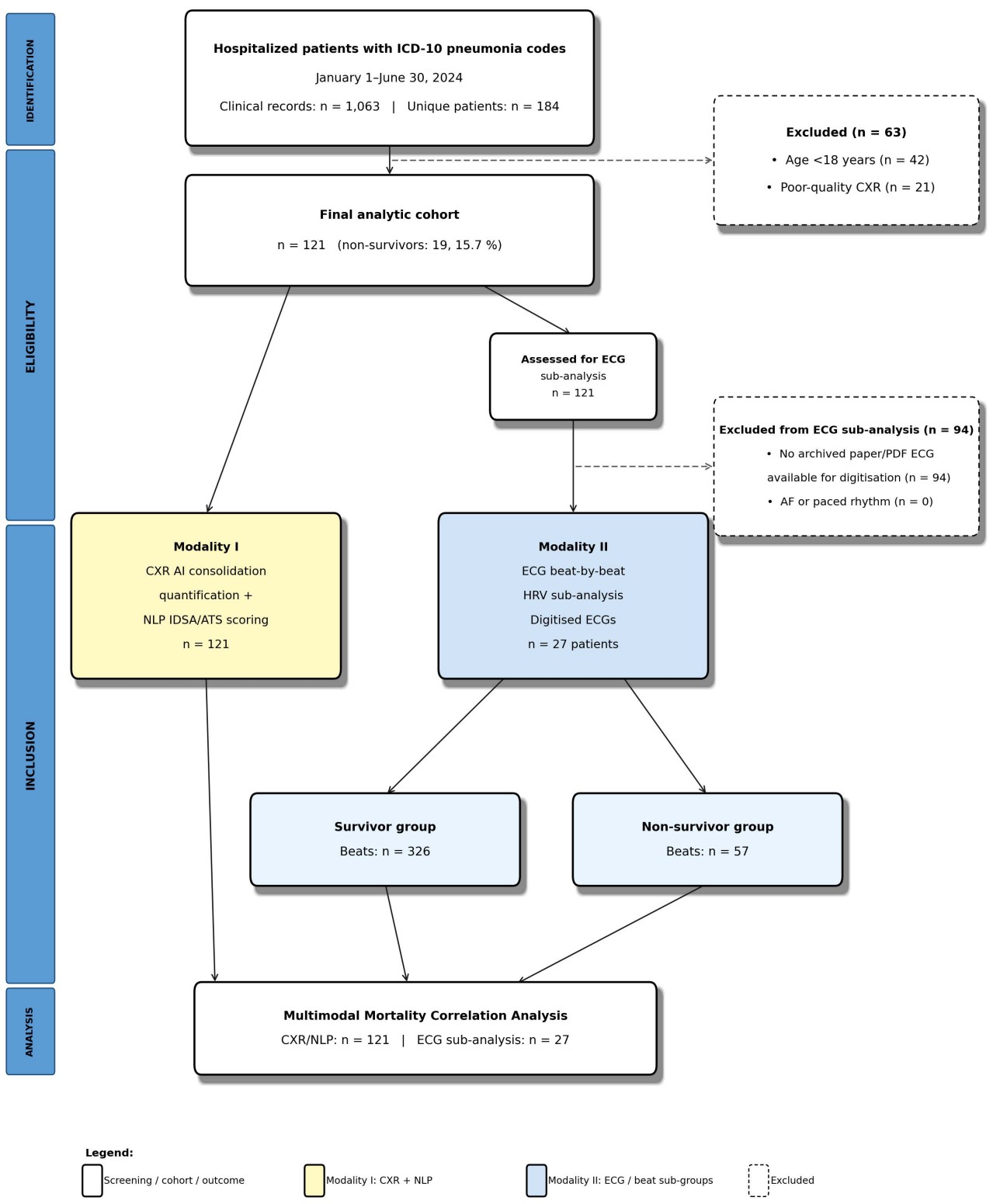

**Fig 1. Participant flow diagram.** Patients admitted January 1 to June 30, 2024 with ICD-10 pneumonia codes were screened (n = 184). After exclusions (age < 18 years, n = 42; poor-quality chest radiograph, n = 21), 121 patients comprised the final analytic cohort. All 121 patients underwent NLP-based IDSA/ATS severity scoring and AI-based CXR lung consolidation quantification. A subset of 27 patients with available digitised paper-based ECGs

underwent beat-by-beat ECG/HRV analysis (survivor beats n = 326; non-survivor beats n = 57; after tachycardia-overlap correction and multi-lead quality filtering); none were excluded for atrial fibrillation or paced rhythm. All three modalities contributed to the mortality correlation analysis. ICD-10: International Classification of Diseases, 10th revision; CXR: chest anteroposterior radiograph; ECG: electrocardiogram; NLP: natural language processing; AI: artificial intelligence; HRV: heart rate variability.

17.1% (12/70) for males (*p* = 0.797). Age-stratified mortality rates were 12.3% (8/65) for patients ≤65 years and 19.6% (11/56) for patients >65 years (*p* = 0.354). The median length of hospital stay was 11.0 days [IQR 7.0–21.0] overall, showing no significant difference between groups (12.0 days for non-survivors vs. 11.0 days for survivors, *p* = 0.817). Among non-survivors, the median time from admission to death was 12.0 days [IQR 5.0–21.5]. Bacterial pneumonia was the most frequent primary diagnosis (40.5% overall; 47.4% in non-survivors, 39.2% in survivors). No statistically significant differences were identified in the distribution of primary diagnoses, number of antibiotics used (6.0 vs. 4.0, *p* = 0.102), or duration of antibiotic treatment (6.1 vs. 6.0 days, *p* = 0.292) between groups (Table 1).

### NLP and CXR AI-derived multimodal biomarkers

Analysis using NLP and CXR AI-derived variables identified several factors significantly associated with mortality (Fig 2). A principal diagnosis of pneumonia due to *Klebsiella pneumoniae* showed the strongest association with increased mortality (odds ratio [OR] 18.94, 95% confidence interval [CI] 1.85–193.44). Higher total lung compromise ratio (per standard deviation [SD] increase) was also significantly associated with increased mortality (OR 8.32, 95% CI 1.23–56.29). A higher IDSA/ATS severity score derived via NLP (per SD increase) was associated with increased mortality risk (OR 1.78, 95% CI 1.09–2.88), and the AI classification confidence score was likewise significant (OR 1.69, 95% CI 1.02–2.79). A general diagnosis of pneumonia showed a trend toward significance (OR 3.62, 95% CI 0.94–13.87, *p* = 0.070). Other factors, including individual bilateral lung compromise ratios (right: OR 1.36, *p* = 0.23; left: OR 0.84, *p* = 0.50), NLP-extracted age, sex, and other principal diagnoses, did not reach statistical significance.

**Table 1. Clinical characteristics of patients included in the study cohort (n = 121).**

| Variable | Missing | Overall (n = 121) | Non-survivor (n = 19) | Survivor (n = 102) | *p*-value |
|---|---|---|---|---|---|
| Age (years), median [Q1, Q3] | 0 | 66.0 [56.0, 81.0] | 66.0 [56.0, 79.5] | 66.0 [56.0, 79.8] | 0.789 |
| Sex, female, n (%) | 0 | 51 (42.1) | 7 (36.8) | 44 (43.1) | 0.797 |
| Length of stay (days), median [Q1, Q3] | 0 | 11.0 [7.0, 21.0] | 12.0 [5.0, 21.5] | 11.0 [7.2, 21.0] | 0.817 |
| **Primary diagnosis, n (%)** | | | | | |
| Bacterial pneumonia | | 49 (40.5) | 9 (47.4) | 40 (39.2) | 0.615 |
| Viral pneumonia (influenza) | | 14 (11.6) | 2 (10.5) | 12 (11.8) | |
| Diabetes mellitus | | 8 (6.6) | 1 (5.3) | 7 (6.9) | |
| Pleural disease | | 5 (4.1) | 2 (10.5) | 3 (2.9) | |
| Oncological | | 4 (3.3) | 0 | 4 (3.9) | |
| Other diagnoses | | 41 (33.9) | 5 (26.3) | 36 (35.3) | |
| Number of antibiotics used, median [Q1, Q3] | 0 | 4.0 [2.0, 7.0] | 6.0 [3.5, 7.5] | 4.0 [2.0, 7.0] | 0.102 |
| Mean antibiotic duration (days), median [Q1, Q3] | 4 | 6.0 [5.0, 7.0] | 6.1 [5.9, 6.9] | 6.0 [5.0, 7.0] | 0.292 |
| Discharge status (death), n (%) | | 19 (15.7) | 19 (100.0) | 0 (0.0) | <0.001 |

Q1: first quartile; Q3: third quartile. *p*-values: Mann-Whitney U test for continuous variables; Chi-squared or Fisher exact test for categorical variables. "Other diagnoses" category aggregates all primary diagnoses each occurring in ≤3 patients (fractures, lithiasis, other infections, urinary tract infection, peptic ulcer, aspiration pneumonia, arrhythmias, heart failure, cerebrovascular disease, chronic obstructive pulmonary disease (COPD), lung abscess, ischaemic heart disease, and others).

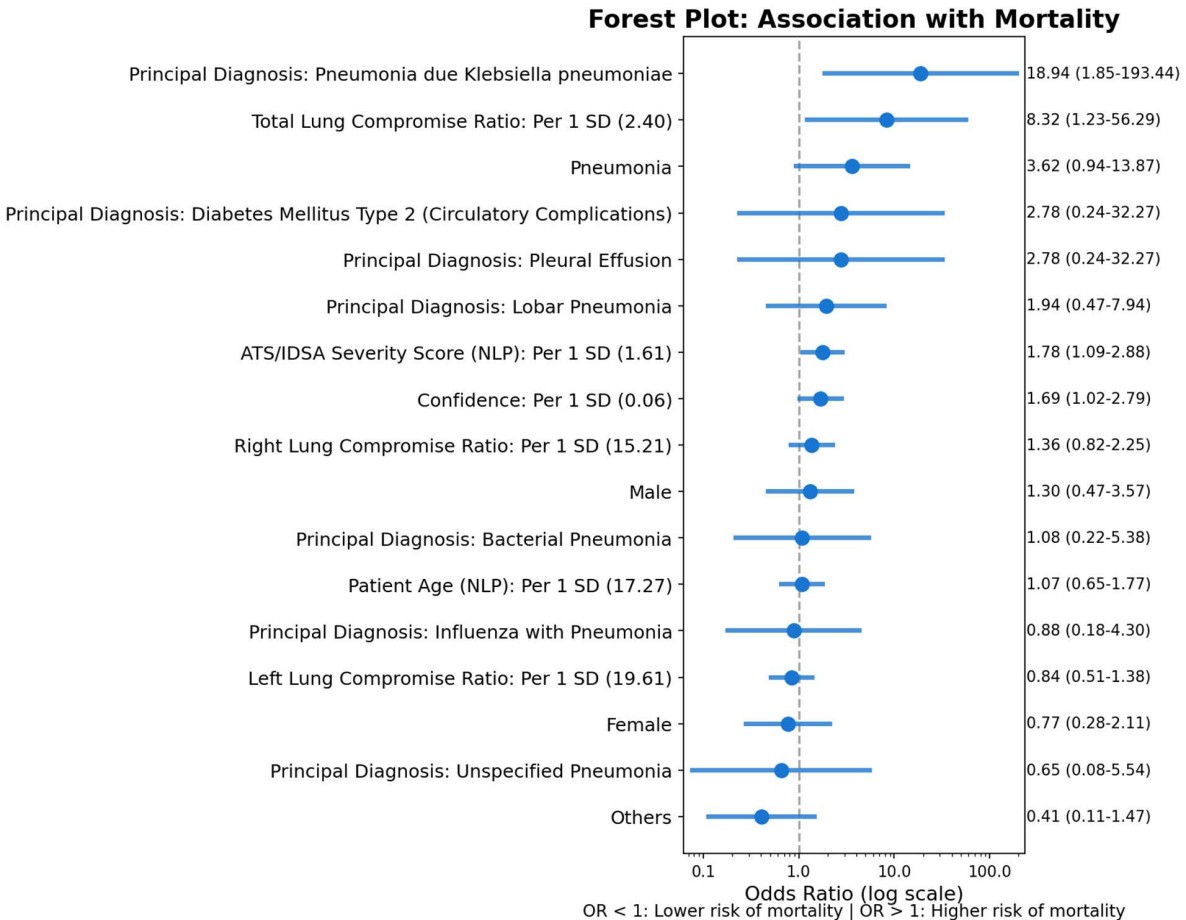

**Fig 2. Forest plot of mortality association with NLP-extracted and AI-quantified CXR features.** Odds ratios (ORs) with 95% confidence intervals (CIs) from separate simple logistic regression models. Continuous variables are expressed per standard deviation increase. Reference category for binary variables is absent/no diagnosis.

### ECG quantitative biomarkers

The analysis of ECG features revealed further associations with mortality (Figs 3, 4, 5). A linear mixed model (LMM) with beats nested within patients assessed 17 morphological features simultaneously. The strongest fixed effects were shorter preceding and following RR intervals in non-survivors (RR$_{prev}$: $\beta$ = -1.34, $p_{raw}$ = 0.008; RR$_{next}$: $\beta$ = -1.33, $p_{raw}$ = 0.008), consistent with tachycardia. All other morphological features (T-wave, ST-segment, QRS, P-wave amplitudes) showed non-significant $\beta$-coefficients ($p_{raw}$>0.4) in the multivariable model.

In a secondary analysis applying Benjamini-Hochberg false discovery rate (BH-FDR) correction to the ECG beat-level linear mixed model (17 features, 383 beats nested in 27 patients), no feature survived correction at $q$<0.05. The strongest signals were RR-interval features (RR$_{prev}$: $\beta$ = -1.34, $p_{raw}$ = 0.008, $q$=0.068; RR$_{next}$: $\beta$ = -1.33, $p_{raw}$ = 0.008, $q$=0.068); all other ECG waveform amplitude features had $q$>0.90. These null FDR-corrected results are consistent with the small exploratory cohort and reinforce that the bivariate associations reported above should be considered hypothesis-generating.

A complementary patient-level analysis of ultra-short (10-second) HRV metrics from the standardised lead II segment (n = 27 patients) revealed clinically coherent group differences (S6 Fig). Non-survivors exhibited higher median heart rate (120.1 vs. 84.4 bpm, $p$ = 0.023, $r$ = -0.83) and lower short-term variability: RMSSD (4.1 vs. 23.5 ms, $p$ = 0.041, $r$ = 0.75) and

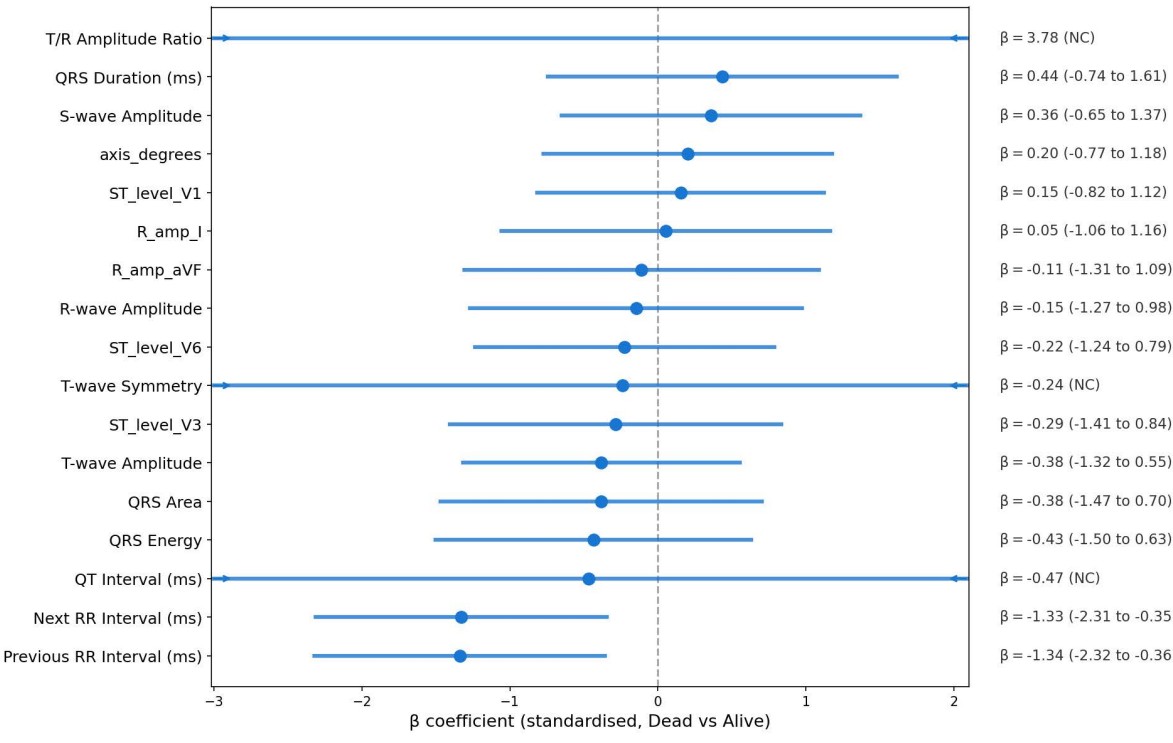

β < 0: Lower in non-survivors | β > 0: Higher in non-survivors | NC: model did not converge

**Fig 3. Forest plot of ECG feature $\beta$-coefficients from the linear mixed model.** Standardised $\beta$-coefficients with 95% confidence intervals from a linear mixed model (mortality as fixed effect, patient as random intercept; 383 beats nested in 27 patients). Features are ordered by effect size. All amplitudes in millivolts (mV); RR intervals in milliseconds. BH-FDR correction applied across 17 features; no feature survived at $q<0.05$ (strongest: $RR_{prev}$ $q=0.068$).

Poincaré SD1 (3.0 vs. 17.6 ms, $p=0.041$, $r=0.75$). SD2 and SDNN showed non-significant trends in the same direction ($p=0.083$). These HRV findings are consistent with reports of reduced parasympathetic modulation in critically ill sepsis patients, where RMSSD and SD1 are depressed relative to survivors [15,16,21].

Lead-by-lead Wilcoxon comparisons across 13 leads further localised the repolarisation signal (S7 Fig): T-wave amplitude was significantly lower in non-survivors in leads II ($q=0.036$, $r=-0.51$), III ($q=0.036$, $r=-0.50$), and V5 ($q=0.036$, $r=-0.54$); ST-segment level was also lower in the long lead II strip ($q<0.001$, $r=-0.42$). All four associations survived BH-FDR correction at $q<0.05$ (39 comparisons).

The average heartbeat waveforms for each group show broad visual similarity across the P wave, QRS complex, ST segment, and T wave (Fig 4). However, quantitative analysis revealed significant differences. Heartbeats from non-survivors exhibited significantly lower median ST-segment values (median -0.053 [IQR -0.078 to -0.042] mV) compared with survivors (median -0.034 [IQR -0.058 to -0.015] mV, $p<0.001$), whereas R-wave amplitudes did not differ significantly between groups (0.337 [0.301–0.354] mV vs. 0.295 [0.181–0.446] mV, $p=0.271$). Full beat-level feature descriptives are provided in S8 Table.

## NLP agreement analysis

The agreement analysis of IDSA/ATS criteria, assessed by two independent internists (inter-human $\kappa$ = 0.408; 95% CI: 0.223–0.592) and the AI models (NLP-V1 vs. human $\kappa$ = 0.510; 95% CI: 0.336–0.684; NLP-V2 vs. human $\kappa$ = 0.455; 95% CI: 0.275–0.635), revealed varying levels of consensus across criteria (Fig 6, Table 2).

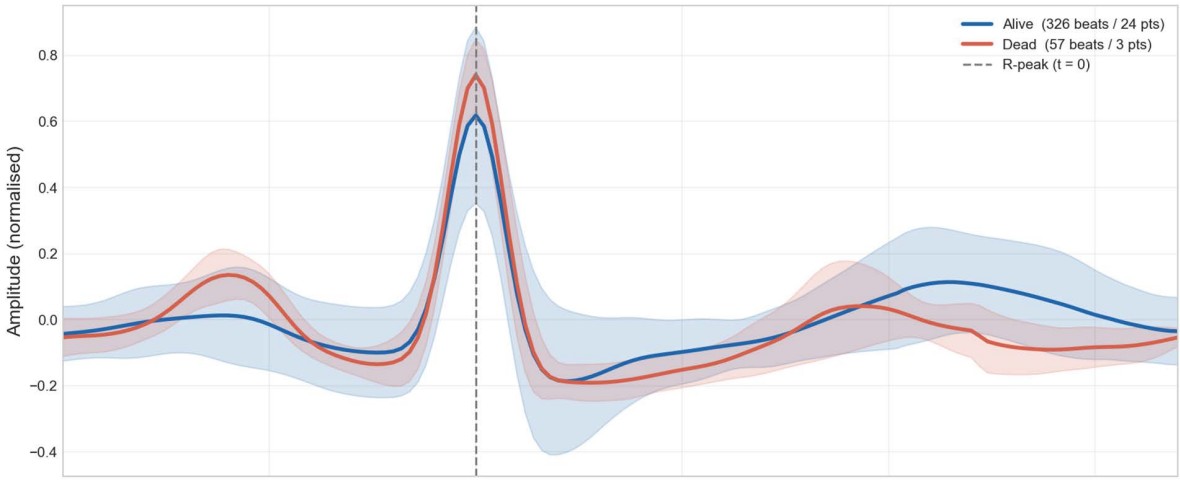

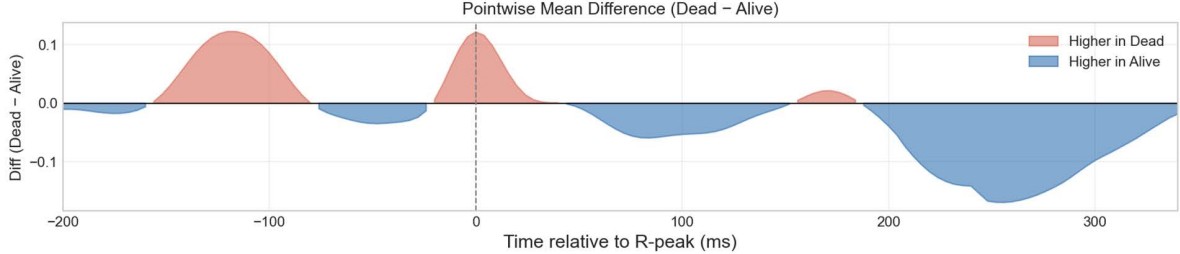

**Fig 4. Detailed beat-by-beat analysis by outcome.** Average heartbeat waveforms for survivors (n = 326 beats) and non-survivors (n = 57 beats) overlaid with ±1 standard deviation. Absolute difference between group means shown in lower panel.

AI models showed mixed performance. For $PaO_2/FiO_2$, multilobar opacities, and uremia, both model versions yielded low agreement (51–69%) with 24–28 false negatives, identifying these as targets for pipeline refinement. Hypotension and leukopenia criteria showed the highest AI agreement (≥90%). Across all criteria, no false positives were recorded, consistent with the conservative regex design in which an absent keyword cannot generate a false positive activation. Per-criterion confusion matrices are shown in S1 Fig; overall agreement statistics in S2 Table; and per-criterion detection rates in S3 Table.

The accumulated false negatives across criteria shifted the NLP-derived IDSA/ATS severity scores downward relative to the human-assigned reference. Comparison of score distributions (S4 Table) indicated that approximately 15–20% of patients were reclassified to a lower risk stratum when relying on NLP labels alone. Despite this attenuation, the NLP-derived severity score retained a statistically significant association with mortality (OR 1.78), suggesting that even a conservatively biased NLP signal contributes prognostic information.

## Discussion

This exploratory study demonstrated the feasibility of automatically deriving candidate mortality biomarkers from multi-modal clinical data (CXR, NLP of Spanish clinical notes, digitised paper ECGs) routinely collected at a tertiary LMIC hospital. The associations identified are consistent with established physiological knowledge and support further investigation.

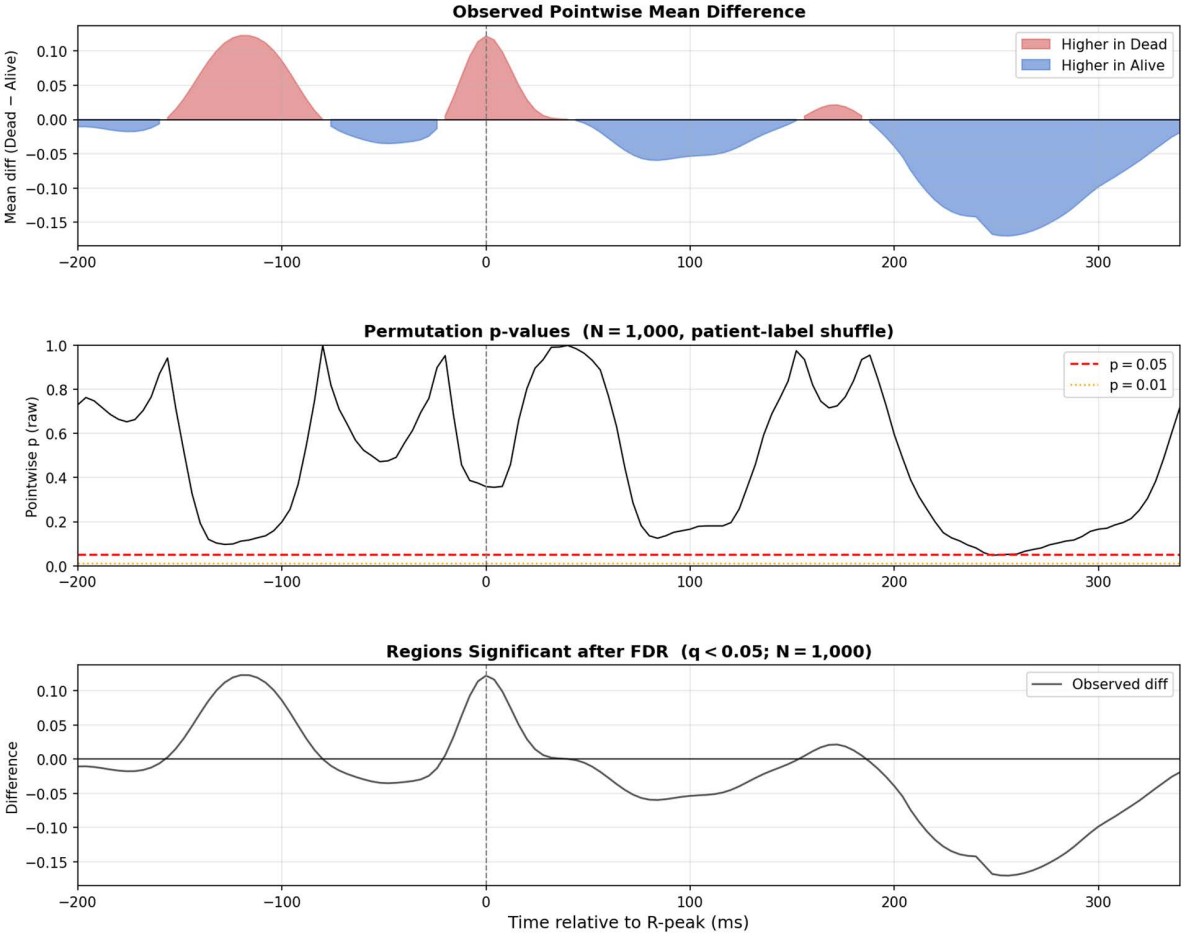

**Fig 5. Pointwise permutation test of beat-template amplitude differences by mortality outcome.** Observed mean difference (Dead - Alive) at each time sample from -200 to +340 ms relative to R-peak, with patient-label permutation p-values (N = 1,000 permutations; FDR-adjusted).

The identification of *Klebsiella pneumoniae* infection via NLP as the strongest predictor of mortality (OR 18.94) aligns with extensive evidence documenting its association with severe outcomes in critically ill patients [22]. Reported mortality rates for *K. pneumoniae* bacteraemia are substantial (29% at 30 days), with odds ratios for death ranging from 3 to 5 for carbapenem-resistant strains [23]. The magnitude of the point estimate in this study is notably high and accompanied by a very wide confidence interval (95% CI 1.85–193.44), a direct consequence of small sample size and low event count (n = 19) [24].

The finding that higher total lung compromise ratio predicted increased mortality (OR 8.32 per SD; 95% CI 1.23–56.29) is consistent with numerous studies demonstrating that radiographic lung involvement correlates with pneumonia severity and adverse outcomes [25]. AI-driven quantification offers a potential advantage of objectivity and reproducibility over manual scoring [26]. Similar to the *Klebsiella* finding, the wide confidence interval reflects the statistical uncertainty inherent in the small sample.

The association between a higher NLP-derived IDSA/ATS severity score and increased mortality (OR 1.78 per SD; 95% CI 1.09–2.88) validates the utility of the NLP component. The IDSA/ATS criteria are well-established tools for risk

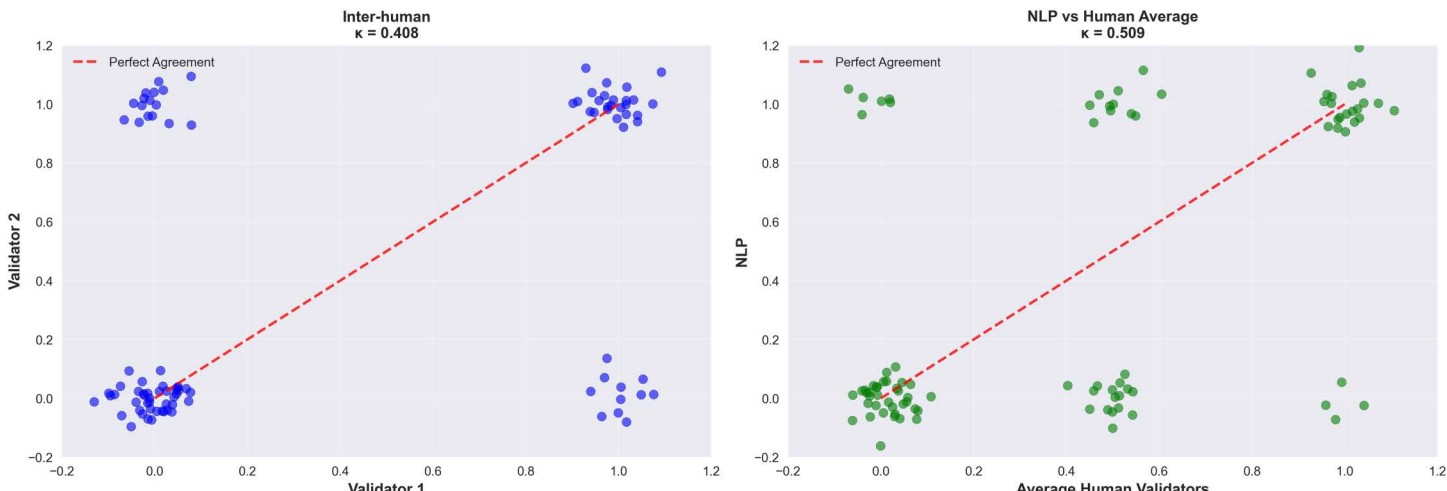

**Fig 6. Agreement analysis of IDSA/ATS criteria by human experts and AI models (V1 and V2).** NLP-V1=dictionary-only regular expressions; NLP-V2=dictionary plus context-window rules. FP=false positives; FN=false negatives. $\kappa$: Cohen's kappa coefficient. Per-criterion confusion matrices are shown in S1 Fig; agreement statistics in S2 Table.

**Table 2. Agreement analysis of IDSA/ATS criteria by human experts and AI models (NLP-V1 and V2). Human agreement reflects proportion of cases where two independent internists reached the same scoring decision. AI agreement measured against human consensus. All False Positives=0 across all criteria (conservative regex design: keyword absence⇒FN, keyword match⇒no FP possible).**

| Criterion | Human Agmt (%) | Human Disagree (n/N) | AI-V1 Agmt (%) | AI-V2 Agmt (%) | AI FP (n) | AI FN (n) |
|---|---|---|---|---|---|---|
| Invasive ventilation | 87.2 | 12/94 | 73.4 | 81.9 | 0 | 15 |
| Septic shock | 80.9 | 18/94 | 89.4 | 89.4 | 0 | 1 |
| Tachypnea | 77.7 | 21/94 | 76.6 | 96.8 | 0 | 2 |
| $PaO_2/FiO_2$ | 61.7 | 36/94 | 55.3 | 55.3 | 0 | 24 |
| Multilobar opacities | 76.6 | 22/94 | 69.1 | 54.3 | 0 | 25 |
| Confusion | 67.0 | 31/94 | 72.3 | 79.8 | 0 | 7 |
| Uremia | 76.6 | 22/94 | 66.0 | 51.1 | 0 | 28 |
| Leukopenia | 91.5 | 8/94 | 97.9 | 91.5 | 0 | 1 |
| Hypotension | 91.5 | 8/94 | 98.9 | 90.4 | 0 | 1 |

Agmt: agreement; FP: false positives; FN: false negatives; NLP: natural language processing; IDSA/ATS: Infectious Diseases Society of America / American Thoracic Society. N=94 for all criteria (subset of cohort with double human annotation).

stratification in community-acquired pneumonia (CAP), with reported AUC 0.88 for mortality prediction; modified versions have shown even higher accuracy (AUC 0.925) [27,28].

Neither the individual bilateral lung compromise ratios (right: OR 1.36, $p=0.23$; left: OR 0.84, $p=0.50$) reached statistical significance, whereas the total compromise ratio aggregating both lungs did (OR 8.32). This suggests that overall parenchymal involvement, rather than lateralised consolidation, drives the mortality association. The bilateral segmentation avoids the quadrant-level artefacts inherent in finer-grained subdivisions that overlap with the cardiac silhouette in standard anteroposterior CXRs [29–31]. External validation with larger cohorts remains necessary before biological interpretation is drawn [32].

In the beat-level LMM, only RR-interval features approached significance ($q=0.068$), consistent with tachycardia as a marker of physiological stress in acute illness [33]. However, the complementary lead-by-lead analysis confirmed that

T-wave amplitude was significantly reduced in non-survivors across leads II, III, and V5 (all $q<0.04$), consistent with literature linking T-wave flattening to adverse outcomes [34]; and ST-segment level was depressed in the continuous lead II strip ($q<0.001$), consistent with ST depression predicting worse outcomes in critically ill patients [35]. Collectively, the convergence of tachycardia (LMM and HRV), depressed short-term parasympathetic modulation (RMSSD, SD1), and spatially corroborated repolarisation abnormalities (T-wave and ST-segment across multiple leads) suggests the pipeline captures a coherent cardiopulmonary stress signature consistent with the severe systemic impact of critical pneumonia.

The ultra-short HRV analysis yielded results directionally consistent with prior sepsis cohorts. Junior et al. reported higher heart rates (120±19 vs. 97±13 bpm, $P=0.005$) and reduced RMSSD (7.5±4.7 vs. 14.6±7.6 ms) and SD1 (5.1±2.9 vs. 9.5±5.7 ms, $P=0.02$) in ICU sepsis non-survivors (n=60) [21]; our findings mirror this pattern with comparable effect sizes (HR: 120.1 vs. 84.4 bpm, $p=0.023$; RMSSD: 4.1 vs. 23.5 ms, $p=0.041$; SD1: 3.0 vs. 17.6 ms, $p=0.041$). A systematic review of HRV as a predictor of sepsis mortality found pooled evidence for reduced time-domain indices in non-survivors, although heterogeneity was high [15]; our observations align with the predominant direction of effect. Papaioannou et al. further established the association between reduced HRV and elevated inflammatory markers in septic ICU patients [36], providing a plausible pathophysiological link for the present findings.

The reliability of HRV metrics derived from ultra-short 10-second ECG recordings warrants discussion. Nussinovitch et al. demonstrated that RMSSD from 10-second segments showed excellent agreement with 5-minute recordings (ICC ≈ 0.91) in healthy volunteers [37] and in patients with diabetes mellitus [38], supporting its use as a time-efficient index of parasympathetic activity. However, SDNN showed insufficient reliability from 10-second windows [39], consistent with our finding that SDNN did not reach significance ($p=0.083$), and suggesting that time-domain metrics other than RMSSD should be interpreted with caution in ultra-short recordings. The multi-lead analysis confirming T-wave amplitude reductions in leads II, III, and V5, together with ST-segment depression in the continuous lead II strip (S7 Fig), provides spatial corroboration of the repolarisation abnormalities identified in the average beat-template analysis.

## Methodological considerations

*Architectural and explainability choices.* The choice of ResNet-18 as the CXR backbone was driven by the resource-constrained deployment target: its ~11.7 million parameters enable inference on a single CPU in ~200 ms per image, making it feasible for LMIC hospitals without GPU infrastructure. This architecture shares the same family as CheXNet [18], an established benchmark for CXR classification [40]. More recent architectures such as Vision Transformers [41] may offer performance improvements but require significantly larger labelled datasets and computational budgets. Future work should benchmark ResNet-18 against these alternatives on CXR severity tasks.

CAM was selected as the saliency method because of its compatibility with the global average pooling layer of ResNet-18 and its computational simplicity (single forward pass). Alternative methods—Grad-CAM [42], Grad-CAM++ [43], SHAP [44], and LIME [45]—offer finer-grained or model-agnostic explanations but at higher computational cost. A systematic comparison of saliency methods applied to CXR consolidation quantification is warranted.

*Computational requirements.* The full multimodal pipeline processes a single patient record (one CXR, one ECG PDF, and associated clinical notes) in approximately 5–10 seconds on a standard quad-core CPU (Intel Core i5, 16 GB RAM) without GPU acceleration. The CXR module accounts for the majority of processing time (~3–5 s for CLAHE preprocessing, lung segmentation, classification, and CAM generation); the NLP regex module runs in <0.5 s; and ECG digitisation with HRV analysis requires ~1–3 s. Integration into Electronic Health Record (EHR) systems would require a REST API wrapper around the processing pipeline, real-time triggering upon document upload, and appropriate clinical decision support display interfaces.

*CURB-65 and PSI benchmarks.* This study used the IDSA/ATS severity criteria as the clinical benchmark for NLP extraction rather than CURB-65 or PSI scores, because several variables required for CURB-65 (blood urea nitrogen, respiratory rate, systolic blood pressure) and PSI (arterial pH, pleural effusion, nursing home residence) were not

systematically documented in the unstructured clinical notes of this retrospective cohort [46]. Future prospective studies should include structured capture of CURB-65/PSI components to enable direct score comparisons.

*False negative impact on severity scoring.* The conservative regex NLP design (zero false positives, non-zero false negatives) inevitably underestimates some patients' IDSA/ATS scores. Analysis of the score distributions under NLP-extracted vs. human-assigned labels (S4 Table) indicated that approximately 15–20% of patients were reclassified to a lower risk stratum when relying on NLP labels alone. This underestimation does not invalidate the mortality association (which was statistically significant despite the attenuation) but suggests that NLP refinement, particularly for $PaO_2/FiO_2$ and multilobar opacities criteria, could strengthen the signal.

*Cross-domain perspective.* The pipeline of digitising analogue physiological recordings, extracting quantitative features, and associating them with clinical outcomes can benefit from advances in other fields. A recent comprehensive survey of ECG analysis using transformer-based architectures [47] highlights the potential for self-supervised pre-training on large unlabelled ECG corpora—an approach particularly relevant for LMIC settings where labelled data is scarce. Future iterations of the ECG module should evaluate these architectures.

*Time-to-event considerations.* The primary analysis used logistic regression to model binary in-hospital mortality rather than time-to-event methods. A Cox proportional hazards sensitivity analysis was not performed because: (a) only 19 events occurred, yielding insufficient power for a semi-parametric survival model with multiple covariates; (b) the competing risk of hospital discharge was not formally modelled; and (c) exact dates of death within the hospitalisation were available for only a subset of non-survivors. The median time from admission to death among non-survivors (12.0 days [IQR 5.0–21.5]) was similar to the median length of stay for survivors (11.0 days [IQR 7.2–21.0]), suggesting that mortality was not concentrated in the early or late phase of hospitalisation. Future prospective studies with larger cohorts should employ Cox regression and competing-risk models to capture the temporal dynamics of mortality risk.

## Limitations and strengths

Strengths of this work include its novel multimodal approach, use of AI modules applicable to data formats common in LMIC settings, focus on a critical clinical outcome, and application of all three modules to the same cohort for joint association analysis.

Major limitations include: (i) single-centre retrospective design with small sample size (n = 121) and low event count (n = 19; events-per-variable ≈ 4 for a 5-predictor model), increasing risks of overfitting and statistical instability; (ii) absence of multiple-testing correction for the primary bivariate analysis; when BH-FDR correction was applied to the ECG beat-level linear mixed model, no feature survived at $q<0.05$ (strongest: RR-interval $q = 0.068$), although complementary HRV and lead-by-lead analyses identified significant group differences (S6 Fig and S7 Fig), underscoring that most individual associations should be interpreted cautiously; (iii) reliance on retrospective EHR data, introducing potential quality issues from missing data and documentation variability; (iv) ECG signals were digitised from PDF archives—native machine-output digital ECG recordings (XML or SCP-ECG format) were not available for any patient in this retrospective cohort, as all ECGs were archived as printed paper reports, precluding a formal Bland-Altman validation against machine output, a limitation common to retrospective LMIC studies [48]; (v) no sensitivity analysis stratifying results by CXR projection type (PA vs. AP/portable) was performed, and saliency robustness (e.g., Grad-CAM++ comparison) was not evaluated; (vi) the regex-based NLP pipeline has inherent limitations compared with deep NLP models: it cannot resolve abbreviation ambiguity (e.g., "SOB"—shortness of breath vs. sobbing), negation contexts (e.g., "no evidence of consolidation"), or cultural and institutional variation in medical Spanish terminology across Latin American countries [20,49]. Regex was chosen over machine learning NLP (TF-IDF, word embeddings, or transformer-based approaches) because the small corpus size (94 annotated notes) was insufficient for supervised training, and the domain-specific terminology favoured a knowledge-driven approach; (vii) the study was conducted at a single centre during a six-month window (January–June 2024) and enrolled a demographically constrained population (median age 66 years, single Colombian city), limiting both

temporal and geographic generalisability; and (viii) although sex was recorded as a binary variable (male/female) and included in the analysis, no formal sex/gender audit per SAGER guidelines [50] was performed, and sex-stratified or age-stratified mortality sub-analyses were not conducted due to the small sample size.

Future research must prioritise rigorous external validation across diverse populations to assess generalisability and fairness. Specifically, a multicentre prospective study enrolling ≥500 patients across at least three hospitals (ideally spanning different Latin American countries) is needed to provide adequate statistical power (EPV ≥ 10) and assess geographic generalisability. Prospective blinded randomised clinical trials are necessary to evaluate real-world impact on clinical decision-making and patient outcomes. The pipeline should be benchmarked against established scoring systems (CURB-65, PSI) in structured data, and the CXR module should be compared with modern architectures (Vision Transformers, attention-based models) using larger training datasets. Sex-stratified and age-stratified sub-analyses should be included to evaluate fairness and equity. Finally, upgrading the NLP module from regex to a Spanish-language clinical language model (e.g., fine-tuned transformer) could substantially reduce false negatives and improve severity score accuracy.

## Conclusion

Given the persistent challenge of high pneumonia mortality and the limitations of current prognostic tools, this exploratory proof-of-concept study demonstrated the feasibility of integrating AI to derive candidate mortality biomarkers from multimodal data—chest radiographs, clinical text, and electrocardiograms—as routinely collected in a resource-limited LMIC hospital. Associations were identified between in-hospital mortality and AI-quantified total lung compromise, NLP-derived IDSA/ATS severity score, and a coherent profile of quantitative ECG/HRV alterations reflecting cardiopulmonary stress. While these multimodal AI results must be interpreted as preliminary and hypothesis-generating, they provide proof-of-concept evidence and compelling hypotheses for larger validation studies and blinded randomised clinical trials, in the continuing fight against pneumonia, the "Captain of the Men of Death" [51].

## Materials and methods

### Ethics statement

The study was approved by the Comité de Ética at Hospital Alma Máter de Antioquia, Medellín, Colombia (approval number IN82–2021). The study used retrospective clinical data collected as part of routine care; the ethics committee granted a waiver of informed consent consistent with national regulations for retrospective observational research. The study followed the STROBE checklist for observational studies (S1 Appendix).

### Data sources and model characteristics

This study utilised existing data originally collected for clinical care, which were subsequently repurposed for model development and validation. Importantly, the three modalities followed distinct analytical paradigms: the CXR module employed deep learning (ResNet-18 pre-training and fine-tuning for classification and CAM-based quantification), whereas the ECG and NLP modules used traditional feature extraction (signal processing with NeuroKit2 and regular-expression pattern matching, respectively) to derive quantitative features for downstream statistical analysis. For initial training of the CXR AI model, several publicly available, anonymised datasets were used: CheXNet [18], Chest X-ray Images [52], Montgomery County chest X-ray set (NIH, 2014) [53], Shenzhen chest X-ray set [54], RSNA Pneumonia Detection Challenge [55], ChestX-ray14 [55], the Open Source COVIDx dataset, and Chest Xray Pneumonia [52]. No synthetic data were generated.

CXR images (n = 395) from Clínica CES (a tertiary hospital in Medellín, January–December 2020) were used for fine-tuning. Images were included if they met the following quality criteria: anteroposterior (AP) or posteroanterior (PA) projection, absence of significant rotation or truncation precluding bilateral lung visualisation, and sufficient contrast for

digital analysis (assessed visually). No minimum pixel resolution was imposed; all clinical DICOM images met adequate resolution for the 224×224 px input required by ResNet-18. Images underwent preprocessing with Contrast Limited Adaptive Histogram Equalisation (CLAHE) [56] and lung segmentation using a DeepLabV3-based model [57]. A ResNet-18 architecture, pre-trained on ImageNet and fine-tuned on public and local datasets, was used for three-class classification (Normal, Pneumonia, Other Disease; ROC curves and calibration in S3 Fig). Model interpretability and lung injury quantification were provided by Class Activation Mapping (CAM) [58]. ResNet-18 was selected as the backbone architecture because its relatively low parameter count (~11.7 million) permits CPU-based inference, it shares the same architectural family as CheXNet [18] (a seminal CXR deep learning model [40]), and it has been widely validated for medical image classification. Modern architectures such as Vision Transformers (ViT) [41] offer potential performance gains but require substantially larger training datasets and computational resources not available in this LMIC context. CAM was chosen over alternative explainability methods—Grad-CAM [42], Grad-CAM++ [43], SHAP [44], and LIME [45]—because of its direct compatibility with the global average pooling layer of ResNet-18, computational simplicity (single forward pass), and established use in CXR literature [18,58].

The CAM output was used to derive consolidation compromise ratios. The segmented lung field was split into right and left lungs along the midline of the segmentation mask (S5 Fig). The compromise ratio for each lung $L$ was defined as:

$$CR_L = \frac{\text{Number of CAM-activated pixels in lung } L}{\text{Total pixels in lung } L} \times 100$$

(1)

where a pixel was considered activated if its CAM value exceeded the threshold $T^*$. The total compromise ratio (TCR) was the ratio of all CAM-activated pixels to total lung pixels (see Fig 7). The optimal CAM activation threshold ($T^* = 0.20$) was determined by leave-one-out cross-validation (LOOCV) against a radiologist ground truth of 106 images [59] (Dice = 0.286±0.199; AUROC = 0.613±0.223; S5 Table, S8 Fig). Full CXR pipeline configuration is documented in S6 Table.

The primary analytic dataset was derived from participants treated at Hospital Alma Máter de Antioquia. This dataset comprised EHR unstructured clinical notes, initial ECG recordings (obtained as PDF images), and first CXR.

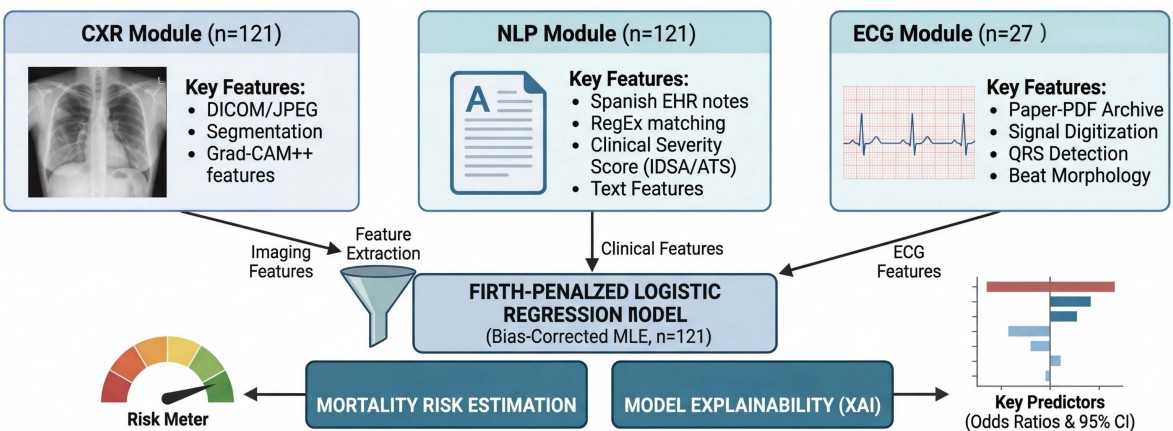

**Fig 7. Multimodal AI pipeline for pneumonia mortality prediction with a representative patient example.** Three parallel analysis modules—CXR (ResNet-18 + Grad-CAM consolidation quantification, $n = 121$), NLP (regex-based IDSA/ATS severity scoring from Spanish-language clinical notes, $n = 121$), and ECG (signal digitisation from paper PDF archives with beat morphology and HRV feature extraction, $n = 27$)—feed extracted features into a Firth penalised logistic regression model for in-hospital mortality risk estimation with explainable feature attribution. Bottom panel: illustrative data flow for a representative non-survivor patient showing CXR compromise ratios, NLP-extracted IDSA/ATS criteria, and ECG-derived biomarkers.

PLOS Digital Health

## ECG digitisation and signal processing

ECG recordings, initially in PDF format, were digitised using a semi-automated process involving scale calibration and baseline detection. Signal processing assumed standard clinical ECG paper speed (25 mm/s) and amplitude calibration (10 mm/mV), yielding an effective sampling resolution of 500 Hz after pixel-to- time interpolation from lead II of the 12-lead ECG. A band-pass filter (0.5–40 Hz, NeuroKit2 default Butterworth parameters) was applied. R-peak detection used the Pan-Tompkins algorithm as implemented in NeuroKit2 [60]; beats with RR intervals outside the physiological range (300–2000 ms) were excluded as artefacts. Patients with atrial fibrillation or artificially paced rhythms identified on visual inspection were excluded from HRV analysis (none met these criteria in the final ECG cohort; Fig 1). Heart rate variability (HRV) analysis was subsequently performed using NeuroKit2 [60] to compute time-domain (MeanNN, SDNN, RMSSD) and non-linear (SD1, SD2, ApEn) metrics. All ECG amplitudes are reported in millivolts (mV); the isoelectric baseline was estimated as the TP-segment mean. Native machine-output digital ECG recordings (XML or SCP-ECG format) were not available for this retrospective cohort—all ECGs were archived as printed paper reports, precluding a formal Bland-Altman comparison against machine output (available digitisation accuracy plots are shown in S2 Fig).

## NLP pipeline

Extracted EHR text underwent standard preprocessing (lowercasing, accent normalisation, and removal of non-semantic tokens). A custom severity scoring algorithm quantified clinical status based on modified IDSA/ATS criteria [27]: a library of predefined clinical terms and associated severity weights in Spanish was represented as regular expressions and applied to the preprocessed text. Positive matches incremented the total severity score. The NLP pipeline used dictionary-based regular expressions augmented with context-window rules to reduce false negatives for compound clinical expressions. An age-based score component was added from the structured EHR fields.

Each record was independently evaluated by two board-certified internists for IDSA/ATS criterion scoring (n = 94 records double-annotated; Cohen's $\kappa$ reported in S2 Table). The conservative regex design implies that a criterion key-word that is absent from a note yields a false negative (FN), not a false positive (FP), explaining the zero-FP result across all criteria in Table 2. Anonymised extraction examples from Spanish clinical notes are provided in S2 Appendix.

## Study participants

Participants were adults retrospectively identified from the primary hospital dataset (January 1 to June 30, 2024). Inclusion criteria: age ≥18 years; primary diagnosis of acute pneumonia confirmed clinically and/or radiologically; and requirement for continuous monitoring (i.e., intensive, intermediate, or step-down care). Exclusion criteria: explicit documented refusal of intubation; and >20% missingness across key variables (S1 Table), a threshold consistent with recommendations for handling incomplete data in medical research [61]. All three modalities (CXR, clinical notes, and ECG where available) were acquired during the index hospital admission, typically within ±24 hours of presentation. Final cohort: n = 121. A sub-set of 27 patients with available paper-based ECGs underwent the ECG/HRV sub-analysis (see Fig 1).

## Statistical analysis

To examine the association between individual features and mortality, a series of univariate and bivariate statistical analyses were conducted using Python (version 3.12) with the SciPy, statsmodels, and NeuroKit2 libraries. Continuous variable normality was assessed using the Shapiro-Wilk or D'Agostino-Pearson test. Differences between survivor and non-survivor groups were evaluated using independent-samples $t$-tests (normally distributed data) or Mann-Whitney U tests (non-normally distributed data); for categorical variables, Chi-squared or Fisher's exact test was used. Odds ratios (ORs) with 95% confidence intervals (CIs) were calculated for each feature independently using simple logistic regression; ORs for continuous variables are expressed per standard deviation increase. A conventional threshold of $p<0.05$ was used;

because this is an exploratory proof-of-concept study with a limited event count (n = 19; events-per-variable ≈ 4), no formal multivariable model was pre-registered as a primary analysis. Given the class imbalance (19 deaths vs. 102 survivors, 15.7% event rate), association measures were based on odds ratios from logistic regression rather than accuracy-based metrics, and exact tests (Fisher's exact test) were preferred for sparse categorical comparisons, consistent with recommendations for rare-event studies [62]. Results were visualised as forest plots. As a sensitivity analysis, a Firth penalised logistic regression was fitted in a separate analysis notebook using three predictors (one per modality: total lung compromise ratio, NLP-IDSA score, and tachycardia-corrected mean heart rate from 10-second Lead II); results are reported in S7 Table, S4 Fig, and S9 Table. Restricted cubic splines or fractional polynomials were considered to capture potential non-linear effects of heart rate and NN interval on mortality; however, the small event count (n = 19) provided insufficient degrees of freedom to estimate non-linear terms reliably, and the analysis was therefore restricted to linear-in-the-logit associations.

## Supporting information

**S1 Fig. NLP per-criterion confusion matrices.** Confusion matrices for each IDSA/ATS criterion comparing NLP-V1 and NLP-V2 predictions against human expert consensus (n = 94 double-annotated records).
(TIF)

**S2 Fig. ECG digitisation Bland-Altman plots.** Available digitisation accuracy plots from the PDF ECG pipeline. Note: comparison against native machine-output digital ECG (XML/SCP-ECG) was not possible for this cohort; all ECGs were archived as paper reports only.
(TIF)

**S3 Fig. CXR binary classification: ROC curves and calibration plots.** Receiver operating characteristic (ROC) curves for pneumonia vs. healthy (AUC = 0.624) and pneumonia vs. other disease (AUC = 0.556) binary comparisons. Calibration plot shown with Brier score.
(TIF)

**S4 Fig. Firth penalised logistic regression: ROC curve and calibration plot.** Receiver operating characteristic curve (AUC = 0.756, 95% CI: 0.625–0.887) and calibration plot (Brier score = 0.116) for the Firth penalised logistic model fitted on Cohort A (n = 121) with three predictors (CXR total compromise ratio, NLP-IDSA score, age). See S7 Table for coefficient estimates.
(TIF)

**S5 Fig. Lung zone schematic and CAM activation example.** Panel A: Bilateral lung division used to compute the compromise ratio ($CR_L$) for the right and left lungs along the segmentation midline. Panel B: Representative CXR processed through the Humath Curie pipeline showing the original image, Class Activation Map (CAM) heatmap overlay with consolidation area in red, and binary consolidation mask at threshold $T^* = 0.20$.
(TIF)

**S6 Fig. Ultra-short (10-second) HRV metrics by mortality outcome.** Box plots comparing heart rate variability indices derived from the standardised lead II segment (n = 27 patients: 24 survivors, 3 non-survivors).
(TIF)

**S7 Fig. Lead-by-lead ECG feature comparison by mortality outcome.** Wilcoxon rank-sum comparisons of R-wave amplitude, ST-segment level, and T-wave amplitude across 13 leads. Four features survived Benjamini-Hochberg FDR correction at $q < 0.05$.
(TIF)

**S8 Fig. CAM activation threshold LOOCV sweep.** Dice coefficient and AUROC as a function of the CAM activation threshold $T$ across 106 radiologist-annotated images. Optimal threshold $T^* = 0.20$ maximised Dice (0.286±0.199; AUROC = 0.613±0.223).
(TIF)

**S1 Table. Variable missingness by cohort.** Percentage of missing values per variable for Cohort A (CXR/NLP analytic cohort, n = 121) and Cohort B (ECG sub-analysis subset, n = 27). Variables with >20% missingness: `img_total_ratio` (89.3% in Cohort A, reflecting the 27/121 ECG subset), and HRV_LF (11.1% in Cohort B).
(XLSX)

**S2 Table. NLP inter-rater and AI-human agreement (Cohen's kappa coefficients).** Per-criterion $\kappa$ with 95% confidence intervals for: inter-human ($\kappa$ = 0.408), NLP-V1 vs. human ($\kappa$ = 0.510), and NLP-V2 vs. human ($\kappa$ = 0.455).
(XLSX)

**S3 Table. NLP per-criterion detection rates.** Per-criterion detection rate (number and percentage of patients detected) for each IDSA/ATS severity criterion, benchmarked against human expert consensus annotations.
(XLSX)

**S4 Table. IDSA/ATS score distribution: observed NLP labels vs. human ground truth.** Score distributions under two scenarios (NLP-extracted vs. human-assigned labels), with percentage of patients reclassified to a higher or lower risk stratum.
(XLSX)

**S5 Table. CXR pipeline validation metrics.** Dice coefficient ± SD = 0.286±0.199, AUROC ± SD = 0.613±0.223, ICC(2,1) = 0.123 [-0.07, 0.35] against radiologist ground truth (n = 106 images with manual segmentation). CAM activation threshold $T^* = 0.20$ (determined by LOOCV).
(XLSX)

**S6 Table. CXR AI model configuration.** Architecture: ResNet-18, 3-class output (NEUMONIA/OTROS/SANO). Preprocessing: CLAHE, DeepLabV3 lung segmentation. Input: 224 × 224 px. CAM via layer4 activations. Pre-training datasets: CheXpert, NIH ChestX-ray14, Montgomery, Shenzhen, RSNA, COVIDx. Fine-tuning: 395 images, Clínica CES. Training hyperparameters: 300 epochs, AdamW (lr = 0.005), cosine annealing with warm restarts, batch size 512.
(XLSX)

**S7 Table. Firth penalised logistic sensitivity model—Cohort A (n = 121) and Cohort B (n = 27).** Penalised logistic regression (Firth method) fitted with three predictors. Cohort A: CXR total compromise ratio, NLP-IDSA score, age; AUC = 0.756, Brier = 0.116. Cohort B: CXR total compromise ratio, NLP-IDSA score, tachycardia-corrected mean heart rate (10-second Lead II, v4.0); AUC = 0.917, Brier = 0.067.
(XLSX)

**S8 Table. ECG beat-level feature descriptives by survival outcome.** Median [IQR] for 16 ECG morphology and interval features (383 beats from 27 patients, v4.0 tachycardia-corrected). Mann-Whitney U $p$-values reported; beats are not independent (see linear mixed-model results in the main text).
(XLSX)

**S9 Table. Variance inflation factors (VIF) for predictors.** VIF values for Cohort A Firth model predictors: CXR = 1.24, NLP-IDSA = 1.18, Age = 1.37 (all < 2; acceptable). Cohort B predictors (v4.0): CXR = 1.14, NLP-IDSA = 1.17, mean HR = 1.05 (all VIF < 2). HRV panel (v4.0 corrected): RMSSD and SD1 show VIF > $10^4$ reflecting mathematical equivalence (SD1 = RMSSD/$\sqrt{2}$).
(XLSX)

**S1 Appendix. STROBE observational study compliance checklist.** Completed STROBE checklist for prospective/retrospective observational studies (cohort design).
(PDF)

**S2 Appendix. Anonymised NLP extraction examples (Spanish clinical notes).** Three representative, fully anonymised Spanish clinical note excerpts with annotated IDSA/ATS criterion matches from the NLP pipeline.
(PDF)

## Acknowledgments

The authors thank the clinical staff at Hospital Alma Máter de Antioquia for support with data collection and the two board-certified internists who independently annotated the IDSA/ATS criteria.

## Author contributions

**Conceptualization:** Alejandro Hernández-Arango, Juan G Paniagua Castrillon, Julián Rondón-Carvajal, Olga Lucia Quintero Montoya.

**Data curation:** Alejandro Hernández-Arango, Daniel Mejía Arrieta, Juan G Paniagua Castrillon, Julián Rondón-Carvajal, Santiago Álvarez-López, José Julián Garcés Echeverri, Olga Lucia Quintero Montoya.

**Formal analysis:** Alejandro Hernández-Arango, Juan G Paniagua Castrillon, Julián Rondón-Carvajal, Wayner Barrios, Hernán Felipe García Arias, Olga Lucia Quintero Montoya.

**Funding acquisition:** Christian Andrés Díaz León, Juan G Paniagua Castrillon, Jesús Francisco Vargas-Bonilla, Olga Lucia Quintero Montoya.

**Investigation:** Alejandro Hernández-Arango, Juan G Paniagua Castrillon, Julián Rondón-Carvajal, David Restrepo, Santiago Álvarez-López, Hernán Felipe García Arias, José Julián Garcés Echeverri, Carlos Salazar-Martinez, Olga Lucia Quintero Montoya.

**Methodology:** Alejandro Hernández-Arango, Juan G Paniagua Castrillon, Julián Rondón-Carvajal, José Julián Garcés Echeverri, Olga Lucia Quintero Montoya.

**Project administration:** Christian Andrés Díaz León, José Julián Garcés Echeverri, Olga Lucia Quintero Montoya.

**Resources:** Christian Andrés Díaz León, Olga Lucia Quintero Montoya.

**Software:** Christian Andrés Díaz León, Juan G Paniagua Castrillon, Melissa Alejandra Acosta, David Restrepo, Wayner Barrios, Hernán Felipe García Arias, Carlos Salazar-Martinez.

**Supervision:** Alejandro Hernández-Arango, Wayner Barrios, Olga Lucia Quintero Montoya.

**Validation:** Alejandro Hernández-Arango, Daniel Mejía Arrieta, Julián Rondón-Carvajal, David Restrepo, Santiago Álvarez-López.

**Visualization:** Alejandro Hernández-Arango, David Restrepo.

**Writing – original draft:** Alejandro Hernández-Arango, Juan G Paniagua Castrillon, Julián Rondón-Carvajal, Wayner Barrios, Olga Lucia Quintero Montoya.

**Writing – review & editing:** Alejandro Hernández-Arango, Daniel Mejía Arrieta, Juan G Paniagua Castrillon, Julián Rondón-Carvajal, Wayner Barrios, Olga Lucia Quintero Montoya.

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
