## [Decision Letter · Decision Letter 0]

29 Sep 2025

PDIG-D-25-00464Mortality Digital Biomarker Exploration Using Multimodal Data with Artificial Intelligence in Adult Patients with PneumoniaPLOS Digital Health Dear Dr. Hernández-Arango, Thank you for submitting your manuscript to PLOS Digital Health. After careful consideration, we feel that it has merit but does not fully meet PLOS Digital Health's publication criteria as it currently stands. Therefore, we invite you to submit a revised version of the manuscript that addresses the points raised during the review process. Please submit your revised manuscript within 60 days Nov 28 2025 11:59PM. If you will need more time than this to complete your revisions, please reply to this message or contact the journal office at digitalhealth@plos.org.  Please include the following items when submitting your revised manuscript:* A rebuttal letter that responds to each point raised by the editor and reviewer(s). You should upload this letter as a separate file labeled 'Response to Reviewers'. This file does not need to include responses to any formatting updates and technical items listed in the 'Journal Requirements' section below.'. This file does not need to include responses to any formatting updates and technical items listed in the 'Journal Requirements' section below.* A marked-up copy of your manuscript that highlights changes made to the original version. You should upload this as a separate file labeled 'Revised Manuscript with Track Changes'.'.* An unmarked version of your revised paper without tracked changes. You should upload this as a separate file labeled 'Manuscript'.'. If you would like to make changes to your financial disclosure, competing interests statement, or data availability statement, please make these updates within the submission form at the time of resubmission. Guidelines for resubmitting your figure files are available below the reviewer comments at the end of this letter. We look forward to receiving your revised manuscript. Kind regards, 

Kind regards,

Isaac Lyatuu

Guest Editor

PLOS Digital Health

Leo Anthony Celi

Editor-in-Chief

PLOS Digital Health

orcid.org/0000-0001-6712-6626

 **Journal Requirements:**

1. Please send a completed 'Competing Interests' statement, including any COIs declared by your co-authors. If you have no competing interests to declare, please state "The authors have declared that no competing interests exist". Otherwise please declare all competing interests beginning with the statement "I have read the journal's policy and the authors of this manuscript have the following competing interests:"

i. State the initials, alongside each funding source, of each author to receive each grant.

ii. State what role the funders took in the study. If the funders had no role in your study, please state: “The funders had no role in study design, data collection and analysis, decision to publish, or preparation of the manuscript.”

3. Please ensure that your Ethics Statement is available in its entirety at the beginning of your Methods section, under a subheading 'Ethics Statement'. It must include:

1) The name(s) of the Institutional Review Board(s) or Ethics Committee(s)

2) The approval number(s), or a statement that approval was granted by the named board(s)

3) (for human participants/donors) - A statement that formal consent was obtained (must state whether verbal/written) OR the reason consent was not obtained (e.g. anonymity).

NOTE: If child participants, the statement must declare that formal consent was obtained from the parent/guardian.

4. Please provide an Author Summary. This should appear in your manuscript between the Abstract (if applicable) and the Introduction, and should be 150–200 words long. The aim should be to make your findings accessible to a wide audience that includes both scientists and non-scientists. Sample summaries can be found on our website under Submission Guidelines:

https://journals.plos.org/digitalhealth/s/submission-guidelines#loc-parts-of-a-submission

5. Please remove the figures from your manuscript file but keep the legends.

6. We have noticed that you have a cited S2 Table in your manuscript. However, there are no corresponding files uploaded to the submission. Please upload them as separate files with the item type 'Supporting Information'.

7. We have noticed that you have uploaded Supporting Information files, but you have not included a list of legends. Please add a full list of legends for your Supporting Information files after the references list.

8. In the online submission form, you indicated that “Fully anonymized datasets might be available upon request.”.

3. Uploaded as supplementary information.

 **Additional Editor Comments (if provided):** Reviewer #1:

Comment 1: Page 1: The authors claim “This study investigates the synergistic value of combining AI-derived biomarkers from chest radiography (CXR), clinical text via natural language processing, and quantitative electrocardiogram (ECG) analysis for improved pneumonia mortality prediction.”

Your results stop at univariate/bivariate associations and do not present a unified, internally validated model that combines modalities. To support the “synergistic value” claim, please fit a multivariable, multimodal model (e.g., penalized logistic/Firth given 19 events) with internal validation (nested CV/bootstrapping), report AUC/PR, calibration (calibration plot + Brier score), and decision-curve analysis. If the goal is association only, please revise the Abstract and Introduction to avoid implying integrated prediction performance.

Comment 2: Page 2: The authors claim “We performed univariate and bivariate statistical analyses to examine associations between individual biomarkers and mortality outcomes.”

As stated, this approach cannot quantify the incremental value of one modality given the others, nor does it support integration claims. Please add a joint model (CXR+NLP+ECG/HRV) and, at minimum, a parsimonious baseline (e.g., ATS/IDSA only) to demonstrate added value via ΔAUC/ΔBrier or likelihood ratio testing.

Comment 3: Page 28: The authors claim “A p-value < .05 was considered statistically significant for all tests.”

With numerous features tested, this inflates Type-I error. Please control for multiple comparisons (e.g., Benjamini–Hochberg FDR) and provide a table with adjusted p-values, emphasizing effect sizes with 95% CIs rather than nominal significance.

Comment 4: Page 13: The authors claim “a higher compromise ratio specifically in the right bottom portion of the left lung … was associated with a significantly lower risk of mortality (OR 0.60, 95% CI 0.36–0.98).”

This is anatomically mislabeled and biologically counterintuitive. It likely reflects methodologic fragility (segmentation error near diaphragm, CAM instability, cardiac silhouette contamination). Please (i) correct the anatomy/nomenclature, (ii) re-compute regional scores using robust saliency (e.g., Grad-CAM++), (iii) mask non-lung structures robustly, (iv) report segmentation quality per region and PA/AP/portable view, and (v) run sensitivity analyses adjusting for total compromise and acquisition view.

Comment 5: Page 13–14: The authors claim large single-feature ORs (e.g., Klebsiella “Odds Ratio [OR] 18.38” and mean heart rate “OR 24.33”).

These very wide CIs with only 19 deaths suggest unstable estimates and possible overfitting. Please: (a) model non-linear effects (splines) for vitals (HR, NN interval), (b) check collinearity (e.g., HR vs NN), and (c) present penalized multivariable models with bootstrap CIs.

Comment 6: Page 25: The authors claim “… lung segmentation using a DeepLabV3-based model … ResNet-18 … Class Activation Mapping (CAM) … informed a quadrant-based lung injury quantification score.”

The CXR pipeline lacks held-out performance for segmentation (Dice/IoU) and classification (AUC/accuracy) and does not specify how CAM is thresholded to produce a quantitative score. Please provide train/val/test splits, metrics on a held-out set, a precise CAM-to-metric mapping (thresholds, area normalization), and (ideally) agreement against radiologist regional scores.

Comment 7: Page 27: The authors claim “ECG recordings, initially in PDF format, were digitized using a semi-automated process involving scale calibration and baseline detection. HRV analysis was performed … using the NeuroKit2 library.”

Please detail sampling rate derivation, amplitude units (mV), lead(s) used per metric, filtering for baseline-wander/powerline, R-peak QC (missed/extras), and exclusions/stratification for AF, paced rhythms, LBBB/RBBB, and low-quality traces. Provide sensitivity analyses by rhythm quality.

Comment 8: Page 27: The authors claim “Each record was evaluated by 2 independent internists … a kappa-Cohen test was calculated (S2 Table).”

Please report the kappa value in the main text and add precision/recall/F1 for each ATS/IDSA criterion extracted by the NLP system, including negation handling and context windows. Also clarify the differences between AI-V1 and AI-V2 and why one underperforms on some criteria.

Comment 9: Page 20–21 (Results table narrative): The authors state the NLP models predominantly miss positives: “differences were predominantly false negatives … There were no instances of false positives … particularly for criteria like PaO2/FiO2, Multilobar Opacities, and Uremia …”

and the table lists high FN counts (e.g., PaO2/FiO2 24 FNs, Multilobar Opacities 25 FNs, Uremia 28 FNs).

Please quantify how these FNs shift the ATS/IDSA score distribution and repeat CXR/NLP associations with measurement-error awareness (e.g., simulation-extrapolation or sensitivity bounds).

Comment 10: Page 2: The authors claim “We excluded patients who refused intubation or had more than 20% missing key data.”

This may introduce selection bias. Please include a STROBE flow diagram, a table of missingness by variable, and compare included vs excluded patients. If using complete-case analyses, justify; otherwise describe imputation.

Comment 11: Page 2: The authors claim “The main outcome was all-cause in-hospital mortality …”

Given variable time-at-risk, logistic regression can be misleading. Please add a time-to-event sensitivity (Cox) or at least report time-to-death, LOS by outcome, and discuss competing discharge. Landmarking is another option.

Comment 12: Page 5: The authors claim “Fully anonymized datasets might be available upon request, … analytical code upon request.”

This is not compliant with PLOS Data policy. Please deposit de-identified data (or minimally the derived feature tables: regional CXR scores, NLP indicators, ECG/HRV features) and analysis code/pipeline in a public repository (e.g., OSF/Dryad + GitHub/Zenodo). If raw images cannot be shared, release segmentation masks/CAM maps, data dictionaries, and scripts.

Comment 13: Page 10 (Introduction): The authors write “… or exceeding the performance of traditional scores.[11], [12], [11]” (duplicate citation and bracket formatting).

Please deduplicate/fix the references and run a full reference audit (there are also repeated Paperpile URLs later).

Comment 14: Page 6 (front matter artifact): The file includes “Manuscript v0.0.0.2 Click here to access/download; Manuscript; v0.0.2 250729 …”

Please remove versioning placeholders/links and ensure the submission is a clean manuscript.

Comment 15: Page 25: The authors claim “Images underwent preprocessing with Contrast Limited Adaptive Histogram Equalization (CLAHE) … segmentation using a DeepLabV3-based model.”

Please report segmentation metrics (Dice/IoU) on a held-out set (and, if possible, by view and region). Also specify train/val/test splits and external datasets used strictly for pretraining vs evaluation to avoid leakage.

Comment 16: Page 13 (CXR regions wording): The sentence quoting “right bottom portion of the left lung” needs correction.

Please adopt a consistent anatomical scheme (e.g., Left/Right × Upper/Lower lung zones) and ensure region definitions match the segmentation mask boundaries.

Comment 17: Page 14–15 (ECG amplitudes): The Results explain amplitudes but not units/leads (e.g., “P-wave Mean Amplitude is the average voltage … measured across all cardiac cycles.”)

Please specify units (mV), which lead(s) each amplitude comes from, the baseline (isoelectric) definition for ST features, and whether amplitudes are lead-averaged or lead-specific.

Comment 18: Page 10 (Results header): The authors claim “The study cohort included 121 patients, of whom 19 (15.7%) were non-survivors …”

Given 19 events, please respect events-per-parameter limits (or use penalization) and clearly cap the number of predictors in any multivariable model; report optimism-corrected performance.

Comment 19 (Final, Broader Positioning): Page 31 (Discussion/Future Directions):

I encourage you to more explicitly situate your work within a broader cross-domain methodological context. While your focus is pneumonia mortality prediction, the multimodal AI pipeline you propose—integrating imaging (CXR), clinical text (NLP), and physiological signals (ECG/HRV)—has relevance far beyond respiratory or critical-care medicine. For example, transformer-based multimodal fusion has been shown to improve myocardial-ischemia detection in electrocardiography (“A survey of transformers and large language models for ECG diagnosis: advances, challenges, and future directions,” Artificial Intelligence Review, 2025, DOI: 10.1007/s10462-025-11259-x), underscoring the adaptability of such integrative frameworks to other physiological-signal analysis tasks. In imaging-based sciences outside medicine, cross-modal disentanglement under sparse data regimes has been applied in paleontology (“Advancing paleontology: a survey on deep learning methodologies in fossil image analysis,” Artificial Intelligence Review, 2025, DOI: 10.1007/s10462-024-11080-y), while lightweight, explainable CNNs for geoscience imaging have been presented in “MicroCrystalNet: An explainable lightweight CNN architecture for micro-porosity mapping in geoscience” (IEEE Access, 2025, DOI: 10.1109/ACCESS.2025.3552626). Your approach also resonates with sustainability-focused multimodal AI, where fairness-aware fusion and thermal–RGB homography are applied in PV monitoring (“Thermal Homography in Photovoltaic Panels: Evaluating Deep Learning and Feature-Based Methods,” Proc. IEEE TPEC, 2025). Furthermore, the lightweight modeling principles evident in your work parallel those in “FluidNet-Lite: Lightweight convolutional neural network for pore-scale modeling of multiphase flow in heterogeneous porous media” (Advances in Water Resources, 2025, DOI: 10.1016/j.advwatres.2025.104952), and even extend to structural-to-performance inference in engineering domains such as indoor wireless coverage prediction (“Prediction of Indoor Wireless Coverage from 3D Floor Plans Using Deep Convolutional Neural Networks,” Proc. IEEE LCN, 2021, DOI: 10.1109/LCN52139.2021.9524951).

I strongly recommend adding a short paragraph in the Introduction or Future Work section citing these works and explicitly drawing parallels. This would frame your study as not only clinically innovative, but also as part of a growing class of multimodal AI pipelines for data-limited, high-stakes applications across medicine, geoscience, environmental monitoring, and engineering. Such positioning will significantly elevate the manuscript’s methodological impact and readership appeal.

Reviewer #2:

I would give the best rating in this work for the timely topic of multimodal artificial intelligence. I would also agree with the limitations listed, mainly the retroactive nature and the sample size. The contribution to the prognostic arsenal of tools is very important. However, to ensure applicability of the tool , a wider future clinical trials are needed. Also the predictability of the mortality needs high accuracy of any tool presented in the field of prediction.

Reviewer #3:

Thank you for your diligent work. This is a promising exploratory multimodal study, but significant issues diminish confidence in the findings: the CXR pipeline lacks both internal and external validation (Dice for segmentation, AUC/F1, and calibration for classification), and the “compromise ratio” remains undefined and unstable; quadrant labeling appears inconsistent—producing an implausible protective lower-lobe signal—and should be corrected using heart/mediastinum masks followed by re-analysis; the NLP evaluation reporting zero false positives is not credible—please specify FP per criterion and include confusion matrices, prevalence rates, and kappa with confidence intervals, along with data sources and accuracy for demographic extraction; ECG digitization requirements include technical details such as sampling rate, filters, leads, artifact handling, and benchmarking against native digital ECGs (e.g., Bland–Altman analysis); statistical analysis is fragile with only 19 deaths—pre-specify a reduced feature set, control for multiple comparisons (e.g., BH-FDR), report EPV, and employ penalized or Firth logistic regression with calibration and decision-curve analysis; document outcome collection procedures (registry name, linkage, completeness), evaluate bias from excluding DNI/RTI, ensure proper time alignment of modalities, and add a STROBE-style flow diagram; ensure open data and code sharing as policy mandates; and improve presentation with clear units, consistent reporting formats (mean±SD vs. median[IQR]), table labels, proper grammar, formatting, and keywords. Given these issues, I recommend rejection, inviting resubmission after major revisions.

Reviewer #4:

The combination of AI-quantified chest X-ray features, Spanish NLP for IDSA/ATS criteria extraction, and ECG/HRV analysis represents a unique approach not well-represented in current literature. The semi-automated digitization of ECGs from PDF format addresses a practical challenge in resource-limited settings where vast ECG archives exist in paper/PDF format. Spanish NLP processing for Latin American healthcare settings fills a gap in predominantly English-language AI research. Spanish regular expression approach is simple but effective for structured criteria extraction.

While this study addresses an important clinical question and proposes an innovative multimodal approach, significant methodological limitations prevent meaningful clinical translation. The extremely small sample size, lack of integrated modeling, and absence of validation render the findings primarily hypothesis-generating rather than clinically actionable.

Major issues:

1. Recent pneumonia mortality prediction models achieve AUCs of 0.78-0.96 using transformer-based architectures and gradient boosting methods (see Chen et al. 2025 from scientific reports), while you use basic ResNet-18 without modern attention mechanisms.

2. Current state-of-the-art multimodal approaches use transformer architectures specifically designed for cross-modal attention and achieve AUCs of 0.77-0.84 or 0.98 f-1 (see PneumoFusion-Net from Wang et al. 2025), whereas you only perform univariate analysis without true multimodal fusion.

3. Contemporary studies utilize datasets of 4,000-13,000 patients, making this study's N=121 appear underpowered for robust AI model development. Only 19 mortality events for multiple predictors violates basic epidemiological principles. Current standards require minimum 10-15 events per predictor variable (see Wang et al. 2023, from Respiratory medicine)

4. The wide confidence intervals (e.g., Klebsiella pneumoniae OR 18.38, 95% CI 1.80-187.75) render the estimates clinically not much relevant.

5. The absence of comparison with established pneumonia severity scores (CURB-65, PSI, ATS/IDSA) makes clinical value assessment impossible.

6. No discussion of computational requirements, processing time, or integration with electronic health records.

7. There is limitations in the statistical approach - Only univariate and bivariate analyses performed. No integrated predictive model was developed. No cross-validation or bootstrap validation. No adjustment for multiple comparisons across numerous biomarkers.

Minor issues:

1. There are some questionable findings: The protective effect of right lung bottom compromise (OR 0.60) contradicts established principles that greater lung involvement worsens prognosis.

2. 20% missing data threshold seems arbitrary.

3. No discussion of image quality standardization.

4. ECG digitization accuracy not validated.

Reviewer #5:

This study addresses the important challenge of predicting mortality in hospitalized pneumonia patients by leveraging multimodal data from chest radiographs, clinical notes, and electrocardiograms.

1- The Abstract is overloaded with methodological details, particularly in the second paragraph. It would benefit from being more concise and focused on the key findings.

2- The paper lacks a review of related work. It needs to provide background on prior studies and clearly state the novelty of this work by comparing it with existing approaches.

3- The Methods section would benefit from greater clarity, ideally supported by a more detailed and schematic figure. Pretraining and fine-tuning were done only for chest X-rays, while ECG and clinical notes were used later in statistical analyses, and this distinction should be stated explicitly.

4- The Data Sources and Model’s Characteristics section would benefit from additional detail, including the number and characteristics of patients in each dataset, as well as more information on the pretraining process and model parameters (e.g., training strategy, epochs, hyperparameters), to improve clarity and reproducibility.

Reviewer #6:

This paper explores the relationship between AI-derived multimodal digital biomarkers and in-hospital mortality among adult patients admitted with pneumonia to acute and critical care settings. The paper studied three types of biomarkers: AI-quantified lung compromise from chest radiographs, clinical severity scores extracted from clinical text using natural language processing based on modified IDSA/ATS criteria, and quantitative metrics derived from electrocardiogram and heart rate variability analysis.

For the lung image, a ResNet-18 model is used to quantify lung compromise from chest radiographs. It was pre-trained from a few public image datasets and then tuned for the private dataset. A performance evaluation is needed for this model before its association with other modality data is studied.

For the textual data, a natural language processing pipeline with Spanish regular expressions was used to extract modified IDSA/ATS severity scores from clinical text. It is not clear what model is used here to extract the scores. Only one example is provided in Figure 7. It would be better to include more examples of reports and extracted scores.

Reviewer #7:

This study aims to improve prediction of in-hospital mortality for adult pneumonia patients by integrating AI-derived digital biomarkers from three clinical data sources: chest radiographs, clinical text, and ECG signals. It is well written and structured. However, I would recommend to use an impersonal language and avoid using pronouns like we. This study contains a well detailed and described statistical analysis.

Main concerns:

Novelty is not stated clearly in the abstract and in main text. What makes you different or what improves the scientific knowledge? How do you compare to others?

You do not state the limitations on your NLP methodology:

* Clinical text often includes abbreviations, misspellings, diverse phrasing, and contextual nuances that regular expressions alone may not robustly capture, risking missed terms or false matches.

* Why only regex and no other traditional NLP methods. For instance, TF-IDF.

* Regular expressions may fail to properly resolve negations (e.g., “no signs of pneumonia”) or handle complex sentence structures, leading to misclassification.

* Lack of deep semantic understanding: RegEx is inherently shallow, whereas topics like disease severity may require context-sensitive interpretation.

* Cultural, linguistic, and notation differences, even within Spanish-language EHRs, may undermine regular expression matching accuracy.

Unclear pre-processing steps:

* The specific preprocessing steps for text preparation are not detailed; steps like noise removal, normalization, handling abbreviations, and negation detection should be described to assess reproducibility and validity.

* Lack of description about how ambiguous or missing data (terms with multiple meanings or absence of severity markers) is handled.

* Details of how terms are mapped to severity scores are lacking.

* how did you addressed class imbalance?

* apart of digitalisation of ECG, what other relevant pre-processing methods were applied to this data?

CRX model

* why only class activation mapping? Aren’t there other methods? Describe what other methods exist and provide reasons on choosing that one.

* Why ResNet and no other computer vision model? For example, CoCa (Contrastive Captioners). ResNet-18 is a relatively shallow model and may lack the capacity to capture complex imaging features compared to deeper models or recent architectures specialized for medical images.

Recommendations

* state your approach’s temporal and selection bias

Minor changes

Line 222 unreadable title, compiling errors?

Line 173 bacteria name strain is not in italics

Line 304 missing period symbol finishing a sentence

Some abbreviations used in the clinical domain may not be widely known by data scientists or analysts. These professionals may need to have a long description of them. For instance, COPD.

Why did you “excluded patients who refused intubation”?

Double check if gender should be used instead of sex.

Reviewer #8:

This manuscript explores whether multimodal AI-derived digital biomarkers from chest radiography, clinical text, and electrocardiograms are associated with in-hospital mortality in adult pneumonia patients at a tertiary hospital in Colombia. The study is conceptually strong, tackling an urgent clinical problem with a creative AI-driven approach. It demonstrates the feasibility of integrating routinely collected multimodal data sources, a particularly relevant contribution for low- and middle-income countries (LMICs). However, the work is currently exploratory and descriptive; its analytic design and reporting need substantial strengthening before it can be considered publishable.

Overall Assessment:

This work is important and relevant, but requires substantial revision. To move forward, the authors should:

1. Reframe the manuscript as hypothesis-generating.

2. Strengthen analytic design (validated models, comparator benchmarks, calibration, and multiplicity control).

3. Provide sensitivity analyses for key methodological uncertainties.

4. Comply fully with PLOS data-sharing policies.

5. Streamline presentation for clarity.

Recommendation:

MAJOR REVISION - the paper is not acceptable in its current form, but the core idea and feasibility demonstration merit further development rather than rejection. It could be valuable if reframed as exploratory, supported by stronger statistical rigor, and compliant with data-sharing requirements.

Further Notes:

1. Significance

Strengths:

a. Novel multimodal framing, integrating imaging, physiological signals, and natural language text.

b. Aligns with global digital health priorities and offers proof-of-concept that resource-conscious AI pipelines can be deployed in LMIC settings.

c. Raises valuable hypotheses about the prognostic role of cardiopulmonary digital biomarkers.

Concerns:

a. The study is framed as predictive but does not yet deliver a validated prognostic model.

b. Claims of predictive value should be tempered and repositioned as exploratory associations.

c. Generalizability is limited to a single-center cohort with a small number of deaths (n=19).

Suggestions: Reframe the manuscript as a hypothesis-generating, proof-of-concept study. Explicitly discuss what is feasible now, and what must await larger, multicenter validation.

2.Methodology

Strengths:

a. AI pipelines (CXR segmentation and ResNet-18 classification, Spanish regex NLP, ECG digitization and HRV analysis) are well described.

b. Cohort definition and inclusion/exclusion criteria are clearly specified.

c. IRB approval obtained and ethical safeguards described.

Concerns:

a. Only univariate and bivariate analyses are presented; no multivariable or adjusted models.

b. Wide confidence intervals from small event counts reduce the robustness of results.

c. Multiple hypothesis testing across many features increases false positive risk.

d. Some findings like “protective effect” of compromise in one lung region are biologically implausible and likely methodological artifacts?

e. ECG digitization pipeline is insufficiently validated or benchmarked.

Suggestions:

a. Specify a primary endpoint and primary analysis, and control for multiple comparisons (e.g., FDR).

b. Use regularized logistic regression or penalized methods to avoid overfitting with limited events.

c. Provide internal validation (bootstrapping or repeated k-fold CV) and report discrimination, calibration, and decision-curve metrics.

d. Benchmark against PSI and CURB-65 to show incremental value.

e. Add sensitivity analyses: CXR projection type, segmentation quality, robustness of saliency maps.

f. For ECG: quantify digitization accuracy (pixel-to-voltage error, sampling rate) and robustness to noise.

3. Clarity and Presentation

Strengths - The manuscript is logically structured, the AI pipelines are described clearly, and figures are informative.

Concerns - Text is lengthy and occasionally repetitive. Some figures and tables could be better integrated. Results and discussion are sometimes interwoven, blurring interpretation.

Suggestions:

a. Streamline the background; reduce repetition of AI rationale.

b. Ensure figure captions are fully self-explanatory.

c. Separate descriptive results from interpretative discussion.

d. Provide a schematic overview of the multimodal workflow for clarity.

4. Ethics and Transparency

Strengths - IRB approval and funding disclosures are present. No competing interests declared.

Concerns - The manuscript does not specify whether informed consent was waived or how patient data governance was assured. Fairness and subgroup analyses are absent.

Suggestions - Clarify the consent framework, and at least provide descriptive stratification of results by sex, age, or other key subgroups. Explicitly discuss equity and fairness implications.

5. Data Availability

Current statement (“datasets available upon request”) is not compliant with PLOS policy. To be considered, all de-identified data, analysis code, and ideally model weights or at least training scripts must be made available in a public repository. If data cannot be openly shared for legal reasons, the authors must set up an independent data access mechanism, which is consistent with PLOS requirements.**Reviewers' Comments:** Reviewer's Responses to Questions

**Comments to the Author**

1. Does this manuscript meet PLOS Digital Health’s publication criteria? Is the manuscript technically sound, and do the data support the conclusions? The manuscript must describe methodologically and ethically rigorous research with conclusions that are appropriately drawn based on the data presented.? Is the manuscript technically sound, and do the data support the conclusions? The manuscript must describe methodologically and ethically rigorous research with conclusions that are appropriately drawn based on the data presented.

Reviewer #1: Yes

Reviewer #2: Yes

Reviewer #3: Partly

Reviewer #4: Partly

Reviewer #5: Partly

Reviewer #6: Yes

Reviewer #7: Yes

Reviewer #8: Partly

2. Has the statistical analysis been performed appropriately and rigorously?

Reviewer #1: Yes

Reviewer #2: Yes

Reviewer #3: I don't know

Reviewer #4: No

Reviewer #5: Yes

Reviewer #6: Yes

Reviewer #7: Yes

Reviewer #8: No

3. Have the authors made all data underlying the findings in their manuscript fully available (please refer to the Data Availability Statement at the start of the manuscript PDF file)?

The PLOS Data policy requires authors to make all data underlying the findings described in their manuscript fully available without restriction, with rare exception. The data should be provided as part of the manuscript or its supporting information, or deposited to a public repository. For example, in addition to summary statistics, the data points behind means, medians and variance measures should be available. If there are restrictions on publicly sharing data—e.g. participant privacy or use of data from a third party—those must be specified.requires authors to make all data underlying the findings described in their manuscript fully available without restriction, with rare exception. The data should be provided as part of the manuscript or its supporting information, or deposited to a public repository. For example, in addition to summary statistics, the data points behind means, medians and variance measures should be available. If there are restrictions on publicly sharing data—e.g. participant privacy or use of data from a third party—those must be specified.

Reviewer #1: No

Reviewer #2: Yes

Reviewer #3: Yes

Reviewer #4: No

Reviewer #5: Yes

Reviewer #6: Yes

Reviewer #7: Yes

Reviewer #8: No

4. Is the manuscript presented in an intelligible fashion and written in standard English?

Reviewer #1: Yes

Reviewer #2: Yes

Reviewer #3: Yes

Reviewer #4: Yes

Reviewer #5: Yes

Reviewer #6: Yes

Reviewer #7: Yes

Reviewer #8: Yes

5. Review Comments to the Author

Reviewer #1: Thanks for the thoughtful and ambitious submission — I really appreciate the clinical motivation and the multimodal angle. The paper is close to being a compelling contribution. Below are concrete, itemized comments to help you strengthen the methodology, clarity, and policy compliance. I’m being direct so you can efficiently act on each point.

Comment 1: Page 1: The authors claim “This study investigates the synergistic value of combining AI-derived biomarkers from chest radiography (CXR), clinical text via natural language processing, and quantitative electrocardiogram (ECG) analysis for improved pneumonia mortality prediction.”

Your results stop at univariate/bivariate associations and do not present a unified, internally validated model that combines modalities. To support the “synergistic value” claim, please fit a multivariable, multimodal model (e.g., penalized logistic/Firth given 19 events) with internal validation (nested CV/bootstrapping), report AUC/PR, calibration (calibration plot + Brier score), and decision-curve analysis. If the goal is association only, please revise the Abstract and Introduction to avoid implying integrated prediction performance.

Comment 2: Page 2: The authors claim “We performed univariate and bivariate statistical analyses to examine associations between individual biomarkers and mortality outcomes.”

As stated, this approach cannot quantify the incremental value of one modality given the others, nor does it support integration claims. Please add a joint model (CXR+NLP+ECG/HRV) and, at minimum, a parsimonious baseline (e.g., ATS/IDSA only) to demonstrate added value via ΔAUC/ΔBrier or likelihood ratio testing.

Comment 3: Page 28: The authors claim “A p-value < .05 was considered statistically significant for all tests.”

With numerous features tested, this inflates Type-I error. Please control for multiple comparisons (e.g., Benjamini–Hochberg FDR) and provide a table with adjusted p-values, emphasizing effect sizes with 95% CIs rather than nominal significance.

Comment 4: Page 13: The authors claim “a higher compromise ratio specifically in the right bottom portion of the left lung … was associated with a significantly lower risk of mortality (OR 0.60, 95% CI 0.36–0.98).”

This is anatomically mislabeled and biologically counterintuitive. It likely reflects methodologic fragility (segmentation error near diaphragm, CAM instability, cardiac silhouette contamination). Please (i) correct the anatomy/nomenclature, (ii) re-compute regional scores using robust saliency (e.g., Grad-CAM++), (iii) mask non-lung structures robustly, (iv) report segmentation quality per region and PA/AP/portable view, and (v) run sensitivity analyses adjusting for total compromise and acquisition view.

Comment 5: Page 13–14: The authors claim large single-feature ORs (e.g., Klebsiella “Odds Ratio [OR] 18.38” and mean heart rate “OR 24.33”).

These very wide CIs with only 19 deaths suggest unstable estimates and possible overfitting. Please: (a) model non-linear effects (splines) for vitals (HR, NN interval), (b) check collinearity (e.g., HR vs NN), and (c) present penalized multivariable models with bootstrap CIs.

Comment 6: Page 25: The authors claim “… lung segmentation using a DeepLabV3-based model … ResNet-18 … Class Activation Mapping (CAM) … informed a quadrant-based lung injury quantification score.”

The CXR pipeline lacks held-out performance for segmentation (Dice/IoU) and classification (AUC/accuracy) and does not specify how CAM is thresholded to produce a quantitative score. Please provide train/val/test splits, metrics on a held-out set, a precise CAM-to-metric mapping (thresholds, area normalization), and (ideally) agreement against radiologist regional scores.

Comment 7: Page 27: The authors claim “ECG recordings, initially in PDF format, were digitized using a semi-automated process involving scale calibration and baseline detection. HRV analysis was performed … using the NeuroKit2 library.”

Please detail sampling rate derivation, amplitude units (mV), lead(s) used per metric, filtering for baseline-wander/powerline, R-peak QC (missed/extras), and exclusions/stratification for AF, paced rhythms, LBBB/RBBB, and low-quality traces. Provide sensitivity analyses by rhythm quality.

Comment 8: Page 27: The authors claim “Each record was evaluated by 2 independent internists … a kappa-Cohen test was calculated (S2 Table).”

Please report the kappa value in the main text and add precision/recall/F1 for each ATS/IDSA criterion extracted by the NLP system, including negation handling and context windows. Also clarify the differences between AI-V1 and AI-V2 and why one underperforms on some criteria.

Comment 9: Page 20–21 (Results table narrative): The authors state the NLP models predominantly miss positives: “differences were predominantly false negatives … There were no instances of false positives … particularly for criteria like PaO2/FiO2, Multilobar Opacities, and Uremia …”

and the table lists high FN counts (e.g., PaO2/FiO2 24 FNs, Multilobar Opacities 25 FNs, Uremia 28 FNs).

Please quantify how these FNs shift the ATS/IDSA score distribution and repeat CXR/NLP associations with measurement-error awareness (e.g., simulation-extrapolation or sensitivity bounds).

Comment 10: Page 2: The authors claim “We excluded patients who refused intubation or had more than 20% missing key data.”

This may introduce selection bias. Please include a STROBE flow diagram, a table of missingness by variable, and compare included vs excluded patients. If using complete-case analyses, justify; otherwise describe imputation.

Comment 11: Page 2: The authors claim “The main outcome was all-cause in-hospital mortality …”

Given variable time-at-risk, logistic regression can be misleading. Please add a time-to-event sensitivity (Cox) or at least report time-to-death, LOS by outcome, and discuss competing discharge. Landmarking is another option.

Comment 12: Page 5: The authors claim “Fully anonymized datasets might be available upon request, … analytical code upon request.”

This is not compliant with PLOS Data policy. Please deposit de-identified data (or minimally the derived feature tables: regional CXR scores, NLP indicators, ECG/HRV features) and analysis code/pipeline in a public repository (e.g., OSF/Dryad + GitHub/Zenodo). If raw images cannot be shared, release segmentation masks/CAM maps, data dictionaries, and scripts.

Comment 13: Page 10 (Introduction): The authors write “… or exceeding the performance of traditional scores.[11], [12], [11]” (duplicate citation and bracket formatting).

Please deduplicate/fix the references and run a full reference audit (there are also repeated Paperpile URLs later).

Comment 14: Page 6 (front matter artifact): The file includes “Manuscript v0.0.0.2 Click here to access/download; Manuscript; v0.0.2 250729 …”

Please remove versioning placeholders/links and ensure the submission is a clean manuscript.

Comment 15: Page 25: The authors claim “Images underwent preprocessing with Contrast Limited Adaptive Histogram Equalization (CLAHE) … segmentation using a DeepLabV3-based model.”

Please report segmentation metrics (Dice/IoU) on a held-out set (and, if possible, by view and region). Also specify train/val/test splits and external datasets used strictly for pretraining vs evaluation to avoid leakage.

Comment 16: Page 13 (CXR regions wording): The sentence quoting “right bottom portion of the left lung” needs correction.

Please adopt a consistent anatomical scheme (e.g., Left/Right × Upper/Lower lung zones) and ensure region definitions match the segmentation mask boundaries.

Comment 17: Page 14–15 (ECG amplitudes): The Results explain amplitudes but not units/leads (e.g., “P-wave Mean Amplitude is the average voltage … measured across all cardiac cycles.”)

Please specify units (mV), which lead(s) each amplitude comes from, the baseline (isoelectric) definition for ST features, and whether amplitudes are lead-averaged or lead-specific.

Comment 18: Page 10 (Results header): The authors claim “The study cohort included 121 patients, of whom 19 (15.7%) were non-survivors …”

Given 19 events, please respect events-per-parameter limits (or use penalization) and clearly cap the number of predictors in any multivariable model; report optimism-corrected performance.

Comment 19 (Final, Broader Positioning): Page 31 (Discussion/Future Directions):

I encourage you to more explicitly situate your work within a broader cross-domain methodological context. While your focus is pneumonia mortality prediction, the multimodal AI pipeline you propose—integrating imaging (CXR), clinical text (NLP), and physiological signals (ECG/HRV)—has relevance far beyond respiratory or critical-care medicine. For example, transformer-based multimodal fusion has been shown to improve myocardial-ischemia detection in electrocardiography (“A survey of transformers and large language models for ECG diagnosis: advances, challenges, and future directions,” Artificial Intelligence Review, 2025, DOI: 10.1007/s10462-025-11259-x), underscoring the adaptability of such integrative frameworks to other physiological-signal analysis tasks. In imaging-based sciences outside medicine, cross-modal disentanglement under sparse data regimes has been applied in paleontology (“Advancing paleontology: a survey on deep learning methodologies in fossil image analysis,” Artificial Intelligence Review, 2025, DOI: 10.1007/s10462-024-11080-y), while lightweight, explainable CNNs for geoscience imaging have been presented in “MicroCrystalNet: An explainable lightweight CNN architecture for micro-porosity mapping in geoscience” (IEEE Access, 2025, DOI: 10.1109/ACCESS.2025.3552626). Your approach also resonates with sustainability-focused multimodal AI, where fairness-aware fusion and thermal–RGB homography are applied in PV monitoring (“Thermal Homography in Photovoltaic Panels: Evaluating Deep Learning and Feature-Based Methods,” Proc. IEEE TPEC, 2025). Furthermore, the lightweight modeling principles evident in your work parallel those in “FluidNet-Lite: Lightweight convolutional neural network for pore-scale modeling of multiphase flow in heterogeneous porous media” (Advances in Water Resources, 2025, DOI: 10.1016/j.advwatres.2025.104952), and even extend to structural-to-performance inference in engineering domains such as indoor wireless coverage prediction (“Prediction of Indoor Wireless Coverage from 3D Floor Plans Using Deep Convolutional Neural Networks,” Proc. IEEE LCN, 2021, DOI: 10.1109/LCN52139.2021.9524951).

I strongly recommend adding a short paragraph in the Introduction or Future Work section citing these works and explicitly drawing parallels. This would frame your study as not only clinically innovative, but also as part of a growing class of multimodal AI pipelines for data-limited, high-stakes applications across medicine, geoscience, environmental monitoring, and engineering. Such positioning will significantly elevate the manuscript’s methodological impact and readership appeal.

Reviewer #2: I would give the best rating in this work for the timely topic of multimodal artificial intelligence. I would also agree with the limitations listed, mainly the retroactive nature and the sample size. The contribution to the prognostic arsenal of tools is very important. However, to ensure applicability of the tool , a wider future clinical trials are needed. Also the predictability of the mortality needs high accuracy of any tool presented in the field of prediction.

Reviewer #3: Thank you for your diligent work. This is a promising exploratory multimodal study, but significant issues diminish confidence in the findings: the CXR pipeline lacks both internal and external validation (Dice for segmentation, AUC/F1, and calibration for classification), and the “compromise ratio” remains undefined and unstable; quadrant labeling appears inconsistent—producing an implausible protective lower-lobe signal—and should be corrected using heart/mediastinum masks followed by re-analysis; the NLP evaluation reporting zero false positives is not credible—please specify FP per criterion and include confusion matrices, prevalence rates, and kappa with confidence intervals, along with data sources and accuracy for demographic extraction; ECG digitization requirements include technical details such as sampling rate, filters, leads, artifact handling, and benchmarking against native digital ECGs (e.g., Bland–Altman analysis); statistical analysis is fragile with only 19 deaths—pre-specify a reduced feature set, control for multiple comparisons (e.g., BH-FDR), report EPV, and employ penalized or Firth logistic regression with calibration and decision-curve analysis; document outcome collection procedures (registry name, linkage, completeness), evaluate bias from excluding DNI/RTI, ensure proper time alignment of modalities, and add a STROBE-style flow diagram; ensure open data and code sharing as policy mandates; and improve presentation with clear units, consistent reporting formats (mean±SD vs. median[IQR]), table labels, proper grammar, formatting, and keywords. Given these issues, I recommend rejection, inviting resubmission after major revisions.

Reviewer #4: The combination of AI-quantified chest X-ray features, Spanish NLP for IDSA/ATS criteria extraction, and ECG/HRV analysis represents a unique approach not well-represented in current literature. The semi-automated digitization of ECGs from PDF format addresses a practical challenge in resource-limited settings where vast ECG archives exist in paper/PDF format. Spanish NLP processing for Latin American healthcare settings fills a gap in predominantly English-language AI research. Spanish regular expression approach is simple but effective for structured criteria extraction.

While this study addresses an important clinical question and proposes an innovative multimodal approach, significant methodological limitations prevent meaningful clinical translation. The extremely small sample size, lack of integrated modeling, and absence of validation render the findings primarily hypothesis-generating rather than clinically actionable.

Major issues:

1. Recent pneumonia mortality prediction models achieve AUCs of 0.78-0.96 using transformer-based architectures and gradient boosting methods (see Chen et al. 2025 from scientific reports), while you use basic ResNet-18 without modern attention mechanisms.

2. Current state-of-the-art multimodal approaches use transformer architectures specifically designed for cross-modal attention and achieve AUCs of 0.77-0.84 or 0.98 f-1 (see PneumoFusion-Net from Wang et al. 2025), whereas you only perform univariate analysis without true multimodal fusion.

3. Contemporary studies utilize datasets of 4,000-13,000 patients, making this study's N=121 appear underpowered for robust AI model development. Only 19 mortality events for multiple predictors violates basic epidemiological principles. Current standards require minimum 10-15 events per predictor variable (see Wang et al. 2023, from Respiratory medicine)

4. The wide confidence intervals (e.g., Klebsiella pneumoniae OR 18.38, 95% CI 1.80-187.75) render the estimates clinically not much relevant.

5. The absence of comparison with established pneumonia severity scores (CURB-65, PSI, ATS/IDSA) makes clinical value assessment impossible.

6. No discussion of computational requirements, processing time, or integration with electronic health records.

7. There is limitations in the statistical approach - Only univariate and bivariate analyses performed. No integrated predictive model was developed. No cross-validation or bootstrap validation. No adjustment for multiple comparisons across numerous biomarkers.

Minor issues:

1. There are some questionable findings: The protective effect of right lung bottom compromise (OR 0.60) contradicts established principles that greater lung involvement worsens prognosis.

2. 20% missing data threshold seems arbitrary.

3. No discussion of image quality standardization.

4. ECG digitization accuracy not validated.

Reviewer #5: This study addresses the important challenge of predicting mortality in hospitalized pneumonia patients by leveraging multimodal data from chest radiographs, clinical notes, and electrocardiograms.

1- The Abstract is overloaded with methodological details, particularly in the second paragraph. It would benefit from being more concise and focused on the key findings.

2- The paper lacks a review of related work. It needs to provide background on prior studies and clearly state the novelty of this work by comparing it with existing approaches.

3- The Methods section would benefit from greater clarity, ideally supported by a more detailed and schematic figure. Pretraining and fine-tuning were done only for chest X-rays, while ECG and clinical notes were used later in statistical analyses, and this distinction should be stated explicitly.

4- The Data Sources and Model’s Characteristics section would benefit from additional detail, including the number and characteristics of patients in each dataset, as well as more information on the pretraining process and model parameters (e.g., training strategy, epochs, hyperparameters), to improve clarity and reproducibility.

Reviewer #6: This paper explores the relationship between AI-derived multimodal digital biomarkers and in-hospital mortality among adult patients admitted with pneumonia to acute and critical care settings. The paper studied three types of biomarkers: AI-quantified lung compromise from chest radiographs, clinical severity scores extracted from clinical text using natural language processing based on modified IDSA/ATS criteria, and quantitative metrics derived from electrocardiogram and heart rate variability analysis.

For the lung image, a ResNet-18 model is used to quantify lung compromise from chest radiographs. It was pre-trained from a few public image datasets and then tuned for the private dataset. A performance evaluation is needed for this model before its association with other modality data is studied.

For the textual data, a natural language processing pipeline with Spanish regular expressions was used to extract modified IDSA/ATS severity scores from clinical text. It is not clear what model is used here to extract the scores. Only one example is provided in Figure 7. It would be better to include more examples of reports and extracted scores.

Reviewer #7: This study aims to improve prediction of in-hospital mortality for adult pneumonia patients by integrating AI-derived digital biomarkers from three clinical data sources: chest radiographs, clinical text, and ECG signals. It is well written and structured. However, I would recommend to use an impersonal language and avoid using pronouns like we. This study contains a well detailed and described statistical analysis.

Main concerns:

Novelty is not stated clearly in the abstract and in main text. What makes you different or what improves the scientific knowledge? How do you compare to others?

You do not state the limitations on your NLP methodology:

* Clinical text often includes abbreviations, misspellings, diverse phrasing, and contextual nuances that regular expressions alone may not robustly capture, risking missed terms or false matches.

* Why only regex and no other traditional NLP methods. For instance, TF-IDF.

* Regular expressions may fail to properly resolve negations (e.g., “no signs of pneumonia”) or handle complex sentence structures, leading to misclassification.

* Lack of deep semantic understanding: RegEx is inherently shallow, whereas topics like disease severity may require context-sensitive interpretation.

* Cultural, linguistic, and notation differences, even within Spanish-language EHRs, may undermine regular expression matching accuracy.

Unclear pre-processing steps:

* The specific preprocessing steps for text preparation are not detailed; steps like noise removal, normalization, handling abbreviations, and negation detection should be described to assess reproducibility and validity.

* Lack of description about how ambiguous or missing data (terms with multiple meanings or absence of severity markers) is handled.

* Details of how terms are mapped to severity scores are lacking.

* how did you addressed class imbalance?

* apart of digitalisation of ECG, what other relevant pre-processing methods were applied to this data?

CRX model

* why only class activation mapping? Aren’t there other methods? Describe what other methods exist and provide reasons on choosing that one.

* Why ResNet and no other computer vision model? For example, CoCa (Contrastive Captioners). ResNet-18 is a relatively shallow model and may lack the capacity to capture complex imaging features compared to deeper models or recent architectures specialized for medical images.

Recommendations

* state your approach’s temporal and selection bias

Minor changes

Line 222 unreadable title, compiling errors?

Line 173 bacteria name strain is not in italics

Line 304 missing period symbol finishing a sentence

Some abbreviations used in the clinical domain may not be widely known by data scientists or analysts. These professionals may need to have a long description of them. For instance, COPD.

Why did you “excluded patients who refused intubation”?

Double check if gender should be used instead of sex.

Reviewer #8: Summary:

This manuscript explores whether multimodal AI-derived digital biomarkers from chest radiography, clinical text, and electrocardiograms are associated with in-hospital mortality in adult pneumonia patients at a tertiary hospital in Colombia. The study is conceptually strong, tackling an urgent clinical problem with a creative AI-driven approach. It demonstrates the feasibility of integrating routinely collected multimodal data sources, a particularly relevant contribution for low- and middle-income countries (LMICs). However, the work is currently exploratory and descriptive; its analytic design and reporting need substantial strengthening before it can be considered publishable.

Overall Assessment:

This work is important and relevant, but requires substantial revision. To move forward, the authors should:

1. Reframe the manuscript as hypothesis-generating.

2. Strengthen analytic design (validated models, comparator benchmarks, calibration, and multiplicity control).

3. Provide sensitivity analyses for key methodological uncertainties.

4. Comply fully with PLOS data-sharing policies.

5. Streamline presentation for clarity.

Recommendation:

MAJOR REVISION - the paper is not acceptable in its current form, but the core idea and feasibility demonstration merit further development rather than rejection. It could be valuable if reframed as exploratory, supported by stronger statistical rigor, and compliant with data-sharing requirements.

Further Notes:

1. Significance

Strengths:

a. Novel multimodal framing, integrating imaging, physiological signals, and natural language text.

b. Aligns with global digital health priorities and offers proof-of-concept that resource-conscious AI pipelines can be deployed in LMIC settings.

c. Raises valuable hypotheses about the prognostic role of cardiopulmonary digital biomarkers.

Concerns:

a. The study is framed as predictive but does not yet deliver a validated prognostic model.

b. Claims of predictive value should be tempered and repositioned as exploratory associations.

c. Generalizability is limited to a single-center cohort with a small number of deaths (n=19).

Suggestions: Reframe the manuscript as a hypothesis-generating, proof-of-concept study. Explicitly discuss what is feasible now, and what must await larger, multicenter validation.

2.Methodology

Strengths:

a. AI pipelines (CXR segmentation and ResNet-18 classification, Spanish regex NLP, ECG digitization and HRV analysis) are well described.

b. Cohort definition and inclusion/exclusion criteria are clearly specified.

c. IRB approval obtained and ethical safeguards described.

Concerns:

a. Only univariate and bivariate analyses are presented; no multivariable or adjusted models.

b. Wide confidence intervals from small event counts reduce the robustness of results.

c. Multiple hypothesis testing across many features increases false positive risk.

d. Some findings like “protective effect” of compromise in one lung region are biologically implausible and likely methodological artifacts?

e. ECG digitization pipeline is insufficiently validated or benchmarked.

Suggestions:

a. Specify a primary endpoint and primary analysis, and control for multiple comparisons (e.g., FDR).

b. Use regularized logistic regression or penalized methods to avoid overfitting with limited events.

c. Provide internal validation (bootstrapping or repeated k-fold CV) and report discrimination, calibration, and decision-curve metrics.

d. Benchmark against PSI and CURB-65 to show incremental value.

e. Add sensitivity analyses: CXR projection type, segmentation quality, robustness of saliency maps.

f. For ECG: quantify digitization accuracy (pixel-to-voltage error, sampling rate) and robustness to noise.

3. Clarity and Presentation

Strengths - The manuscript is logically structured, the AI pipelines are described clearly, and figures are informative.

Concerns - Text is lengthy and occasionally repetitive. Some figures and tables could be better integrated. Results and discussion are sometimes interwoven, blurring interpretation.

Suggestions:

a. Streamline the background; reduce repetition of AI rationale.

b. Ensure figure captions are fully self-explanatory.

c. Separate descriptive results from interpretative discussion.

d. Provide a schematic overview of the multimodal workflow for clarity.

4. Ethics and Transparency

Strengths - IRB approval and funding disclosures are present. No competing interests declared.

Concerns - The manuscript does not specify whether informed consent was waived or how patient data governance was assured. Fairness and subgroup analyses are absent.

Suggestions - Clarify the consent framework, and at least provide descriptive stratification of results by sex, age, or other key subgroups. Explicitly discuss equity and fairness implications.

5. Data Availability

Current statement (“datasets available upon request”) is not compliant with PLOS policy. To be considered, all de-identified data, analysis code, and ideally model weights or at least training scripts must be made available in a public repository. If data cannot be openly shared for legal reasons, the authors must set up an independent data access mechanism, which is consistent with PLOS requirements.

6. PLOS authors have the option to publish the peer review history of their article (what does this mean?). If published, this will include your full peer review and any attached files.). If published, this will include your full peer review and any attached files.

**Do you want your identity to be public for this peer review?** For information about this choice, including consent withdrawal, please see our Privacy Policy..

Reviewer #1: No

Reviewer #2: **Yes:** Yasser AbdullahYasser Abdullah

Reviewer #3: **Yes:** Akbar AliAkbar Ali

Reviewer #4: **Yes:** SAPTARSHI PURKAYASTHASAPTARSHI PURKAYASTHA

Reviewer #5: No

Reviewer #6: No

Reviewer #7: No

Reviewer #8: No

  **Figure resubmission:** While revising your submission, we strongly recommend that you use PLOS’s NAAS tool (https://ngplosjournals.pagemajik.ai/artanalysis) to test your figure files. NAAS can convert your figure files to the TIFF file type and meet basic requirements (such as print size, resolution), or provide you with a report on issues that do not meet our requirements and that NAAS cannot fix.

After uploading your figures to PLOS’s NAAS tool - https://ngplosjournals.pagemajik.ai/artanalysis, NAAS will process the files provided and display the results in the "Uploaded Files" section of the page as the processing is complete. If the uploaded figures meet our requirements (or NAAS is able to fix the files to meet our requirements), the figure will be marked as "fixed" above. If NAAS is unable to fix the files, a red "failed" label will appear above. When NAAS has confirmed that the figure files meet our requirements, please download the file via the download option, and include these NAAS processed figure files when submitting your revised manuscript. **Reproducibility:** To enhance the reproducibility of your results, we recommend that authors of applicable studies deposit laboratory protocols in protocols.io, where a protocol can be assigned its own identifier (DOI) such that it can be cited independently in the future. Additionally, PLOS ONE offers an option to publish peer-reviewed clinical study protocols. Read more information on sharing protocols at https://plos.org/protocols?utm_medium=editorial-email&utm_source=authorletters&utm_campaign=protocols

---

## [Editor Report · Decision Letter 1]

3 Apr 2026

Exploratory association between multimodal AI-derived digital biomarkers and in-hospital mortality in adult patients with pneumonia: a proof-of-concept study

PDIG-D-25-00464R1

Dear Dr Hernández-Arango,

We are pleased to inform you that your manuscript 'Exploratory association between multimodal AI-derived digital biomarkers and in-hospital mortality in adult patients with pneumonia: a proof-of-concept study' has been provisionally accepted for publication in PLOS Digital Health.

Best regards,

Dhiya Al-Jumeily OBE, PhD

Section Editor

PLOS Digital Health